# Provable Faster Zeroth-order Method for Bilevel Optimization: Achieving Best-known Dependency on Error and Dimension

## Abstract

In this paper, we study and analyze zeroth-order stochastic approximation algorithms for solving black-box bilevel optimization problems, where only the upper and lower function values can be obtained. (Aghasi and Ghadimi, 2024) proposed the first full zeroth-order bilevel method that utilizes Gaussian smoothing to estimate the first- and second-order partial derivatives of functions with two independent blocks of variables. However, this method suffers from a high dimensional dependency of $\mathcal{O}((d_1 + d_2)^4)$, where $d_1$ and $d_2$ are the dimensions of the outer and inner problems, respectively. They left an open question: can this dimension dependency be improved? To answer this question, we propose a single-loop accelerated zeroth-order bilevel algorithm, which achieves a dimension dependency of $\mathcal{O}(d_1 + d_2)$ by incorporating coordinate-wise smoothing gradient estimators (coord). We develop a new theoretical analysis for the proposed algorithm, which converges to a stationary point of $\Phi(x)$ with a complexity of $\mathcal{O}((d_1 + d_2)\epsilon^{-3})$ in expectation settings and $\mathcal{O}((d_1 + d_2)\sqrt{n}\epsilon^{-2})$ in finite sum settings. These complexities are both Best-known with respect to dimension and error $\epsilon$. We also provide experiment to validate the effectiveness of the proposed algorithm.

## 1 Introduction

The goal of bilevel optimization is to minimize the upper-level (UL) function $f(x, y)$ under the constraint that $y$ is minimized with respect to the lower-level (LL) function $g(x, y)$. Bilevel optimization has received increasing attention due to its wide applications in many machine learning problems, including adversarial networks (Goodfellow et al., 2020), hyperparameter tuning (Franceschi et al., 2018), neural architecture search (Liang et al., 2020), meta-learning (Rajeswaran et al., 2019), reinforcement learning (Sutton and Barto, 2018). Formally, it is defined as,

$$\min_{x \in \mathbb{R}^{d_1}} \Phi(x) = f(x, y^*(x)), \quad y^*(x) \in \arg\min_{y \in \mathbb{R}^{d_2}} g(x, y).$$

In this work, we focus on the setting where the lower-level objective $g(x, y)$ is strongly convex in $y$ for any $x$, and the UL objective $f(x, y)$ is possibly non-convex, which is among the most common in this research field. Since deterministic approaches necessitate the evaluation of the full dataset at every iteration, demanding huge computational resources, so it is needed to consider stochastic method. In many applications of interest, the objective functions $f$ and $g$ have the finite-sum form:

$$f(x, y) = \frac{1}{n}\sum_{i=1}^{n} f(x, y; \zeta_i), \ g(x, y) = \frac{1}{n}\sum_{i=1}^{n} g(x, y; \xi_i),$$

where $\zeta_i$ and $\xi_i$ are independent and identically distributed random variables. When dealing potentially infinite number of data samples, $f$ and $g$ are commonly represented as the expectation form:

$$f(x, y) = \mathbb{E}_\zeta[f(x, y; \zeta)], \ g(x, y) = \mathbb{E}_\xi[g(x, y; \xi)].$$

In strongly convex case, the LL problem has unique solution $y^*(x)$, by implicit function theorem, we can obtain explicit form of hypergradient $\nabla\Phi(x)$(Ghadimi and Wang, 2018),

$$\nabla\Phi(x) = \nabla_x f(x, y^*(x)) - \nabla_{xy}^2 g(x, y^*(x))[\nabla_{yy}^2 g(x, y^*(x))]^{-1}\nabla_y f(x, y^*(x)). \tag{1}$$

Since the gradient expression of $\Phi(x)$ involves the inverse of the Hessian matrix, it is challenging to employ gradient-based methods to solve bilevel problems. As a result, previous works often required second-order oracles (Dagr'eou et al., 2022; Ji et al., 2020). Fortunately, recent advances have begun addressing this limitation. For example, (Kwon et al., 2023) utilized a penalty method to avoid the second-order term in the gradient, while (Yang et al., 2023; Huang, 2024) employed finite differences to replace the Hessian-vector product. Despite the significant progress in first-order bilevel methods, there are scenarios where zeroth-order methods are still necessary. For various reasons, such as complexity, lack of access to an accurate model, or computational limitations, there may be no or limited access to the objective gradient. In these situations, the common practice is to approximate the gradient using deterministic or randomized finite difference methods (Shi et al., 2021; Nesterov and Spokoiny, 2017). The scope of application for these methods is vast (see the remainder of this section for a comprehensive summary of applications in machine learning, and refer to (Hu et al., 2024; Zhang et al., 2020) for examples in other areas of engineering). Hence, it is natural to explore zeroth-order bilevel optimization, which requires only function value oracle (or their stochastic samples).

Although zeroth-order methods and bilevel optimization have many applications, only a few papers have considered zeroth-order bilevel optimization. In recent years, (Sow et al., 2021) explored a mixed method that utilizes both first-order and zeroth-order oracles. Additionally, (Aghasi and Ghadimi, 2024) first proposed a fully zeroth-order method via Gaussian smoothing; however, its dependency on the problem dimensions can be $\mathcal{O}((d_1 + d_2)^4)$, making it very expensive in practice. Hence, they left an open question:

> Another important direction of research is exploring the dependency of the sample complexity on the problem dimensions, as well as the existence of alternative approaches with a smaller share of dimensionality.

This paper aims to answer this question. We propose a zeroth-order bilevel optimization algorithm called VRZSBO that can achieve $\mathcal{O}(d_1 + d_2)$ dependency on dimension by incorporating coordinate estimators and finite differences.

Before formally introducing our method, we would like to first present the core issues in bilevel optimization.

### 1.1 KEY PROBLEM: HYPER GRADIENT ESTIMATION

The main challenge in performing gradient descent on $\Phi(x)$ is estimating the inverse Hessian $[\nabla_{yy}^2 g(x, y^*(x))]^{-1}$ in (1). Computing the matrix inverse generally incurs a high computational cost of $\mathcal{O}(d^3)$. To reduce this expense, numerous approaches have been proposed to approximate $\nabla\Phi(x)$ at a lower cost. We divide these categories into three groups (1)Neumann series approach (2)Quadratic auxiliary function approach (3)Penalty method,due to page limitations, more detailed content is deferred to the Appendix.:

### 1.2 RELATED WORK

Among all the work, the most closely related works with ours are (Aghasi and Ghadimi, 2024)(zeroth-order bilevel optimization), (Yang et al., 2023)(use gradient difference), (Chu et al., 2024)(use page). Compared to (Aghasi and Ghadimi, 2024), we improved their dependency of dimension from $\mathcal{O}((d_1 + d_2)^4)$ to $\mathcal{O}(d_1 + d_2)$, and they don't provide analysis of finite sum case. Compared to (Yang et al., 2023), we only use function oracles while they use gradient oracles, and they don't provide analysis of finite sum case. Compared with (Chu et al., 2024), we only use function oracles while they use Hessian-vetor product oracles.

In conclusion our main contribution can be summarized as follows

- By incorporating zeroth-order Coord gradient estimator, we improve zeroth-order complexity dependency of dimension from $\mathcal{O}((d_1 + d_2)^4)$ to $\mathcal{O}(d_1 + d_2)$, which is Best-known in dependence of dimenison, as shown in Table 1.

Table 1: Comparison of different NC-SC bilevel methods for finding an $\epsilon$- stationary point of $\Phi(x)$. Oracle type denotes what kind of oracle the method are use. $\mathcal{C}_L^{a,k}$ denotes $a$-times differentiability with Lipschitz $k$-th order derivatives.

| Method | oracle type | Upper-level $f$ | Lower-level $g$ | Finite Sum | Expectation |
|--------|-------------|-----------------|-----------------|------------|-------------|
| F³SA (Kwon et al., 2023) | 1ˢᵗ | $C_L^{2,2}$ | $C_L^{2,2}$ | – | $\tilde{\mathcal{O}}(\epsilon^{-5})$ |
| AccBO(Gong et al., 2024) | Hv/Jv | $(L_0, L_1)$-smooth | $C_L^{2,2}$ | – | $\tilde{\mathcal{O}}(\epsilon^{-3})$ |
| SPABA(Chu et al., 2024) | Hv/Jv | $C_L^{1,1}$ | $C_L^{2,2}$ | $\mathcal{O}(\sqrt{n}\epsilon^{-2})$ | $\mathcal{O}(\epsilon^{-3})$ |
| ZDSBA(Aghasi and Ghadimi, 2024) | 0ᵗʰ | $C_L^{1,1}$ | $C_L^{2,2}$ | – | $\tilde{\mathcal{O}}((d_1 + d_2)^4\epsilon^{-6})$ |
| SABA(Dagr'eou et al., 2022) | Hv/Jv | $C_L^{2,2}$ | $C_L^{3,3}$ | $\mathcal{O}(n^{2/3}\epsilon^{-2})$ | – |
| SRBA(Dagréou et al., 2024) | Hv/Jv | $C_L^{2,2}$ | $C_L^{3,3}$ | $\mathcal{O}(\sqrt{n}\epsilon^{-2})$ | – |
| **VRZSBO**(this paper) | 0ᵗʰ | $C_L^{1,1}$ | $C_L^{2,2}$ | $\mathcal{O}((d_1 + d_2)\sqrt{n}\epsilon^{-2})$ | $\mathcal{O}((d_1 + d_2)\epsilon^{-3})$ |

- We make a decoupling analysis including both finite sum and expectation case, our analysis separates the errors originating from different sources, which means our analysis can be easily transferred into other analyses.
- Compared with the double-loop zeroth-order algorithm in (Aghasi and Ghadimi, 2024), Our algorithm achieves optimal dependency about $\epsilon$ and dimension in both finite sum and expectation case with a simple single-loop structure, as show in Table 1.
- Our algorithm use only standard assumptions, some additional assumption in recent work, for example $g(x, y)$ is $3^{rd}$ Lipschitz(Dagréou et al., 2024) or bounded $\|y^*(x)\|$(Aghasi and Ghadimi, 2024) are not needed in our paper.

## 2 PRELIMINARIES

**Notation** Throughout the paper, $\| \cdot \|$ denotes the Euclidean norm for vectors, and operator norm for matrices, we use hat line $\hat{\nabla}$ to denote zeroth-order estimator, $\xi$ to denote sample(for example $f(x; \xi)$), we denote vector $c := (x, y)$ and Euclidean ball $\mathbb{B}(r) = \{v \in \mathbb{R}^p : \|v\| \leq r\}$. Hv and Jv denote Hessian-vector and Jacobian-vector Products, respectively, $D$ to denote finite difference operator.

**Assumption 1** (Basic Assumptions). *Let vector $c := (x, y)$, the following results hold:(1) $g(x, y)$ is $\mu$-strongly convex with respect to $y$.(2) $f(c)$ and $g(c)$ are L-smooth with respect to $c$.(3) $\|\nabla_y f(x, y)\| \leq C_f$ for a constant $C_f$.(4) $g(c)$ is $\rho$-Hessian Lipschitz with respect to $c$.(5)$\Phi(x)$ are lower bounded.*

**Assumption 2** (Stochastic Case). *The following assumption for stochastic case*

- *Stochastic functions $f(x, y; \xi)$ and $g(x, y, \xi)$ also satisfy assumption 1.*

- $\mathbb{E}[\|\nabla_{xy}^2 g(x, y; \xi) - \nabla_{xy}^2 g(x, y)\|^2] \leq \sigma^2, \mathbb{E}[\|\nabla_{yy}^2 g(x, y; \xi) - \nabla_{yy}^2 g(x, y)\|^2] \leq \sigma^2,$

- $\mathbb{E}[\|\nabla_y g(x, y; \xi) - \nabla_y g(x, y)\|^2] \leq \sigma^2, \mathbb{E}[\|\nabla_y f(x, y; \xi) - \nabla_y f(x, y)\|^2] \leq \sigma^2,$

- $\mathbb{E}[\|\nabla_x f(x, y; \xi) - \nabla_x f(x, y)\|^2] \leq \sigma^2.$

**Remark 1.** *These kind of assumptions are often used in related research (Yang et al., 2023; Aghasi and Ghadimi, 2024) to achieve the state-of-the-art complexity. Furthermore, to simplify the symbols and improve readability,we use the same Lipschitz constant for both $f(x)$ and $g(x)$, we assume the same variance constant for gradient and second order terms. In additional, to improve readability in some part, we denote $L_{max} = \max\{L, \rho\}$ and $\kappa = \frac{L_{max}}{\mu}$.*

## 3 THE PROPOSED ALGORITHM

### 3.1 GENERAL FRAMEWORK

In this part we discuss the main framework, recall that we introduce the framework that use quadratic auxiliary function in (21) , by minimizing the following auxiliary function, we can get an approxima-

---

**Algorithm 1** Variance reduced Zeroth-order Stochastic Bilevel Optimizer (VRZSBO)

---

Initialization: $\hat{h}_0^x = h_0^x, v_{z,0} = \frac{1}{B}\sum_{i=1}^{B} D\hat{H}_v(t+1;\xi_i) - \hat{\nabla}_y f(x_{t+1}, y_{t+1};\zeta_i), v_{x,0} = \frac{1}{B}\sum_{i=1}^{B} \hat{\nabla}_x f(x_{t+1}, y_{t+1};\zeta_i) - D\hat{J}_v(t+1;\xi_i), v_{y,0} = \frac{1}{B}\sum_{i=1}^{B} \hat{\nabla}_y g(x_{t+1}, y_{t+1};\xi_i)$

**for** $t = 0, 1, \cdots, T-1$ **do**

$\quad y_{t+1} = y_t - \eta_y v_{y,t}$ $\hfill \triangleright v_{y,t}$ defined in (16)

$\quad z_{t+1} = \text{Proj}_{\mathbb{B}(r_z)}(z_t - \eta_z v_{z,t})$ $\hfill \triangleright v_{z,t}$ defined in (15)

$\quad x_{t+1} = x_t - \eta_x v_{x,t}$ $\hfill \triangleright v_{x,t}$ defined in (17)

**end for**

---

tion of $z^* := [\nabla_{yy}^2 g(x, y^*(x))]^{-1} \nabla_y f(x, y^*(x))$:

$$R(x, y, z) = \frac{1}{2}z^T \nabla_{yy}^2 g(x, y)z - z^T \nabla_y f(x, y).$$

A natural idea is to perform gradient descent on $R(x, y, z)$, so we define search direction as follows,

$$h_{y,t} := \nabla_y g(x_t, y_t), \ \ h_{z,t} := \nabla_z R(x_t, y_t, z_t) = \nabla_{yy}^2 g(x_t, y_t)z_t - \nabla_y f(x_t, y_t),$$

and hypergradient approximation as follows,

$$h_{x,t} := \nabla_x f(x_t, y_t) - \nabla_{xy}^2 g(x_t, y_t)z_t, \tag{2}$$

for this approximation of hypergradient, we have the following error bound.

**Lemma 1.** *Under assumption 1, for the error between hypergradient $\Phi(x_t)$ and approximation $h_{x,t}$ in (2), we have the following upper bound,*

$$\|h_{x,t} - \nabla\Phi(x_t)\|^2 \le \mathcal{O}(L_{max}^2 \kappa^2)\|y_t - y^*(x)\|^2 + \mathcal{O}(L_{max}^2)\|z_t - z_t^*\|^2. \tag{3}$$

The detailed proof is given in lemma E.15.

**Main Challenge**: We discussed the main framework in Section 3.1. However, one issue is that this equation contains terms involving gradients and matrix-vector products, which are difficult to compute directly in some practical applications. This motivates us to introduce the zeroth-order method, which only requires function value resources (or their stochastic samples). To address the issue, we design the new ZO-estimators and provide some new analysis tools in Sections 3.2 and 3.3.

### 3.2 ESTIMATE GRADIENT

In this section, we discuss how to approximate the gradient using function values. We begin by defining the following zeroth-order operator that approximates the gradient.

**Definition 1** (Zeroth-order estimator). *We define the following coord zeroth order operator that approximates the gradient, this kind of approach can also be found in (Liu et al., 2018; Ji et al., 2019)*

$$\hat{\nabla}_x f(x, y) := \sum_{\ell=1}^{d_1} \frac{1}{v}\left[f(x + v\mathbf{e}_\ell, y) - f(x, y)\right]\mathbf{e}_\ell,$$

$$\hat{\nabla}_x g(x, y) := \sum_{\ell=1}^{d_1} \frac{1}{v}\left[g(x + v\mathbf{e}_\ell, y) - g(x, y)\right]\mathbf{e}_\ell,$$

$$\hat{\nabla}_y f(x, y) := \sum_{\ell=1}^{d_2} \frac{1}{v}\left[f(x, y + v\mathbf{e}_\ell) - f(x, y)\right]\mathbf{e}_\ell,$$

$$\hat{\nabla}_y g(x, y) := \sum_{\ell=1}^{d_2} \frac{1}{v}\left[g(x, y + v\mathbf{e}_\ell) - g(x, y)\right]\mathbf{e}_\ell.$$

The following lemma shows the error of the zeroth-order gradient estimator for any $L$-smooth function.

**Lemma 2** (Gradient estimate error). *For any $L$-smooth function $p(x)$, its gradient $\nabla p(x)$ and its zeroth-order estimator $\hat{\nabla} p(x)$, we have:*

$$\left\| \hat{\nabla} p(x) - \nabla p(x) \right\|^2 \leq \frac{L^2 d v^2}{4}, \tag{4}$$

*the detailed proof is given in lemma D.1 of Appendix.*

By using lemma 2, we can easily obtain the following corollary, choose sufficiently small smoothing parameter $v$, the gradient estimation error can be bounded by a constant $\delta$.

**Corollary 1.** *With sufficiently small zeroth-order smoothing parameter $v$, for any gradient and gradient estimator appears in our paper , we have*

$$\left\| \hat{\nabla} f(c) - \nabla f(c) \right\| \leq \delta \, and \, \left\| \hat{\nabla} g(c) - \nabla g(c) \right\| \leq \delta, \tag{5}$$

*where $\delta = \frac{L\sqrt{d_1 + d_2} v}{2}, \nabla f(c) = [\nabla_x f(c), \nabla_y f(c)]$ and $\nabla g = [\nabla_x g(c), \nabla_y g(c)]$ with $c = (x, y)$.*

### 3.3 ESTIMATE HESSIAN-VECTOR (HV) AND JACOBIAN-VECTOR(JV) PRODUCTS

Next, we discuss how to approximate the martrix-vector products (Hv and Jv) by function values.

**Definition 2** (Estimate Hv/Jv by zeroth-order estimator). *We define the following zeroth order operator that approximates the Hv/Jv*

- *Hv/Jv :*

$$H_v(t) := \nabla_{yy}^2 g(x_t, y_t) z_t, \, J_v(t) := \nabla_{xy}^2 g(x_t, y_t) z_t. \tag{6}$$

   *In the above term, we define the Hessian/Jacobin vector product that will be used in the computation of section 3.1.*

- *Gradient difference $\overset{approximate}{\Rightarrow}$ Hv/Jv*

$$DH_v(t) := \frac{\nabla_y g(x_t, y_t + h z_t) - \nabla_y g(x_t, y_t)}{h}, \tag{7}$$

$$DJ_v(t) := \frac{\nabla_x g(x_t, y_t + h z_t) - \nabla_x g(x_t, y_t)}{h}. \tag{8}$$

   *In the above term we define an ZO estimator that estimates the Hv term using gradients.*

- *ZO $\overset{approximate}{\Rightarrow}$ gradient difference*

$$D\hat{H}_v(t) := \frac{\hat{\nabla}_y g(x_t, y_t + h z_t) - \hat{\nabla}_y g(x_t, y_t)}{h}, \tag{9}$$

$$D\hat{J}_v(t) := \frac{\hat{\nabla}_x g(x_t, y_t + h z_t) - \hat{\nabla}_x g(x_t, y_t)}{h}. \tag{10}$$

   *In the above term, we define an estimator that use zeroth-order information .*

- *Sampled ZO $\overset{approximate}{\Rightarrow}$ Sampled gradient*

$$D\hat{H}_v(t; \xi) := \frac{\hat{\nabla}_y g(x_t, y_t + h z_t; \xi) - \hat{\nabla}_y g(x_t, y_t; \xi)}{h}, \tag{11}$$

$$D\hat{J}_v(t; \xi) := \frac{\hat{\nabla}_x g(x_t, y_t + h z_t; \xi) - \hat{\nabla}_x g(x_t, y_t; \xi)}{h}. \tag{12}$$

   *In the above term, we define an estimator that use stochastic zeroth-order information .*

*In Definition 2, we define some estimators that estimate gradient difference using function values. Next, we give the following error bound*

**Lemma 3.** *Under assumption 1, and $\left(D\hat{H}_v(t), DH_v(t), D\hat{J}_v(t), J_v(t)\right)$ is given in Definition 2 with $h = \frac{2}{r_z}\sqrt{\frac{\delta}{\rho}}$ where $r_z = \max_{x \in \mathbb{R}^{d_1}} \|z^*(x)\|$, we have*

$$\left\|D\hat{H}_v(t) - H_v(t)\right\|^2 \leq \mathcal{O}(\delta) \, and \, \left\|D\hat{J}_v(t) - J_v(t)\right\|^2 \leq \mathcal{O}(\delta). \tag{13}$$

*The proof sketch in terms of $Hv$ is given as follows:*

$$D\hat{H}_v(t;\xi) \overset{sample}{\Rightarrow} D\hat{H}_v(t) \overset{zeroth\text{-}order}{\underset{lemma \ D.4}{\Rightarrow}} DH_v(t) \overset{finite \ difference}{\underset{lemma \ D.5}{\Rightarrow}} H_v(t),$$

*where lemmas D.4 and D.5 are provided in Appendix. And the similar derivation to $Hv$ for $Jv$ is outlined as follows :*

$$D\hat{J}_v(t;\xi) \overset{approximate}{\Rightarrow} D\hat{J}_v(t) \overset{approximate}{\Rightarrow} DJ_v(t) \overset{approximate}{\Rightarrow} J_v(t).$$

### 3.4 ALGORITHM DESIGN

With the above estimator, we design the following zeroth-order stochastic bilevel optimization algorithm, denoting the zeroth-order approximation from Section 3.2 to Section 3.3 as follows:

$$\hat{h}_{y,t} := \hat{\nabla}_y g(x_t, y_t), \hat{h}_{z,t} := D\hat{H}_v(t) - \hat{\nabla}_y f(x_t, y_t), \hat{h}_{x,t} := \hat{\nabla}_x f(x_t, y_t) - D\hat{J}_v(t). \tag{14}$$

In the finite sum or expectation case, we only have access to a sample point $\xi$. Variance reduction techniques are commonly used to achieve optimal rates in stochastic optimization. Among these, PAGE (Li et al., 2020) is an effective method that utilizes a single-loop iterative format. Drawing from its success, we extend the PAGE concept to our zeroth-order bilevel optimization framework and design the following estimators as stochastic approximations of (14):

$$v_{z,t+1} := \begin{cases} \frac{1}{B}\sum_{i=1}^{B} D\hat{H}_v(t+1;\xi_i) - \hat{\nabla}_y f(x_{t+1}, y_{t+1};\zeta_i) & \text{with probability } p, \\[2ex] v_{z,t} + \frac{1}{b}\sum_{i=1}^{b}\sum D\hat{H}_v(t+1;\xi_i) - D\hat{H}_v(t;\xi_i) & \text{with probability } 1-p. \\ -\frac{1}{b}\sum_{i=1}^{b}\hat{\nabla}_y f(x_{t+1}, y_{t+1};\zeta_i) - \hat{\nabla}_y f(x_t, y_t;\zeta_i) \end{cases} \tag{15}$$

$$v_{y,t+1} := \begin{cases} \frac{1}{B}\sum_{i=1}^{B}\hat{\nabla}_y g(x_{t+1}, y_{t+1};\xi_i) & \text{with probability } p, \\ v_{y,t} + \frac{1}{b}\sum_{i=1}^{b}\hat{\nabla}_y g(x_{t+1}, y_{t+1};\xi_i) - \hat{\nabla}_y g(x_t, y_t;\xi_i) & \text{with probability } 1-p. \end{cases} \tag{16}$$

$$v_{x,t+1} := \begin{cases} \frac{1}{B}\sum_{i=1}^{B}\hat{\nabla}_x f(x_{t+1}, y_{t+1};\zeta_i) - D\hat{J}_v(t+1;\xi_i) & \text{with probability } p, \\[2ex] v_{x,t} + \frac{1}{b}\sum_{i=1}^{b}\hat{\nabla}_x f(x_{t+1}, y_{t+1};\zeta_i) - \hat{\nabla}_x f(x_t, y_t;\zeta_i) & \text{with probability } 1-p. \\ -\frac{1}{b}\sum_{i=1}^{b} D\hat{J}_v(t+1;\xi_i) - D\hat{J}_v(t;\xi_i) \end{cases} \tag{17}$$

Furthermore, our algorithm is formally presented in Algorithm 1.

### 3.5 CONVERGE ANALYSIS

We will analyze the coordinate estimator in both finite-sum and expectation cases. As an example, we provide the key results in the finite-sum setting and include sketch of Theorem 1 in Appendix E.5 to give readers a more intuitive understanding.

**Inner Iteration Analysis**: We first analyze the descent of inner iteration $z_t$ and $y_t$, since the objective function of $z_t$ and $y_t$ are $\mu$-strongly convex and $L$-smooth, we can decrease them linearly as follows.

**Lemma 4.** *Under assumption 1, let $\{z_t\}$ be a sequence generated by Algorithm 1, we have*

$$\left\|z_{t+1} - z^*_{t+1}\right\|^2 \leq (1 - \frac{\eta_z \mu}{6})\|z_t - z^*_t\|^2 + \frac{4\eta_z}{\mu}\left(\left\|v_{z,t} - \hat{h}_{z,t}\right\|^2 + \left\|h_{z,t} - \hat{h}_{z,t}\right\|^2\right)$$

$$- (1 - 2L\eta_z)\|z_{t+1} - z_t\|^2 + \frac{4}{\eta_z \mu}l^2_{Z^*}\|c_{t+1} - c_t\|^2,$$

*where $l_{Z^*}$ represents the smoothness constant of $z^*(x)$. ,and the detailed proof is given in lemma E.6 of Appendix.*

**Lemma 5.** *Under assumption 1, let $\{y_t\}$ be a sequence generated by Algorithm 1, we have*

$$\left\| y_{t+1} - y_{t+1}^* \right\|^2 \leq (1 - \frac{\eta_y \mu}{6}) \left\| y_t - y_t^* \right\|^2 - (1 - 2L\eta_y) \left\| y_{t+1} - y_t \right\|^2 + \frac{4}{\eta_y \mu} l_{Y^*}^2 \left\| x_{t+1} - x_t \right\|^2$$

$$+ \frac{4\eta_y}{\mu} (\left\| v_{y,t} - \hat{h}_{y,t} \right\|^2 + \left\| h_{y,t} - \hat{h}_{y,t} \right\|^2).$$

*where $l_{Y^*}$ represents the smoothness constant of $y^*(x)$, and the proof of this lemma is almost identical to that of lemma 4.*

Next, we need to bound variance terms term $\|v_{y,t} - \hat{h}_{y,t}\|^2$ and $\|v_{z,t} - \hat{h}_{z,t}\|^2$. Firstly, we discuss the variance descent property of $z_t$ in finite sum case

**Lemma 6** (Variance descent in $z$ for finite sum). *Under assumptions 1 and 2, we have*

$$\mathbb{E} \left\| v_{z,t+1} - \hat{h}_{z,t+1} \right\|^2 \leq (1-p)\mathbb{E}[\left\| v_{z,t} - \hat{h}_{z,t} \right\|^2]$$

$$+ \frac{2(1-p)}{b} \left( (8r_z^2 + 3L^2) \left\| c_{t+1} - c_t \right\|^2 + 8L^2 \left\| z_{t+1} - z_t \right\|^2 + 8r_z^2\rho\delta + 6\delta^2 \right).$$

*The detailed proof is given in lemma E.7 of Appendix.*

And variance descent property of $y_t$.

**Lemma 7** (Variance descent in $y$ for finite sum). *Under assumptions 1 and 2, we have*

$$\mathbb{E} \left\| v_{y,t+1} - \hat{h}_{y,t+1} \right\|^2 \leq (1-p)\mathbb{E}[\left\| v_{y,t} - \hat{h}_{y,t} \right\|^2] + \frac{(1-p)}{b} \left( 3L^2 \left\| c_{t+1} - c_t \right\|^2 + 6\delta^2 \right).$$

*The detailed proof is simliar to lemma 6.*

**Outer Iteration Analysis:** For iteration $x_t$, we have similar function value descent lemma and variance descent lemma.

**Lemma 8** (Inexact descent of function $\Phi$). *Under assumptions 1 and 2, let $\{x_t, y_t, z_t\}$ be a sequence generated by Algorithm 1, we have*

$$\Phi(x_{t+1}) \leq \Phi(x_t) - \frac{\eta_x}{2} \left\| \nabla\Phi(x_t) \right\|^2 - \left( \frac{1}{2\eta_x} - \frac{L_\phi}{2} \right) \left\| x_{t+1} - x_t \right\|^2 + \frac{3\eta_x}{2} \left\| v_{x,t} - \hat{h}_{x,t} \right\|^2$$

$$+ \frac{3\eta_x}{2}((2L^2 + 4r_z^2\rho^2) \left\| y_t - y^* \right\|^2 + 4L^2 \left\| z_t - z_t^* \right\|^2 + 2\delta^2 + 8r_z^2\rho\delta),$$

*The detailed proof is given in lemma E.18 .*

Next, we give the descent property of variance term.

**Lemma 9** (Variance descent of $x$ in finite sum case). *Under assumptions 1 and 2, we have*

$$\mathbb{E} \left\| v_{x,t+1} - \hat{h}_{x,t+1} \right\|^2 \leq (1-p)\mathbb{E}[\left\| v_{x,t} - \hat{h}_{x,t} \right\|^2]$$

$$+ \frac{2(1-p)}{b} \left( (8r_z^2 + 3L^2) \left\| c_{t+1} - c_t \right\|^2 + 8L^2 \left\| z_{t+1} - z_t \right\|^2 + 16r_z^2\rho\delta + 6\delta^2 \right).$$

we ref reader to lemma E.19 for detail proof.

Finally by defining the Lyapunov function $\psi_t$:

$$\psi_t := \Phi(x_t) + \frac{18(L^2 + 2r_z^2\rho^2)\eta_x}{\eta_y\mu} \left( \left\| y_t - y_t^* \right\|^2 \right) + \frac{36\eta_x L^2\rho^2}{\eta_z\mu} \left( \left\| z_t - z_t^* \right\|^2 \right)$$

$$+ \frac{3\eta_x}{2p} \left\| v_{x,t} - \hat{h}_{x,t} \right\|^2 + \frac{72(L^2 + 2r_z^2\rho^2)\eta_x}{p\mu^2} \left\| v_{y,t} - \hat{h}_{y,t} \right\|^2 + \frac{144L^2\rho^2\eta_x}{p\mu^2} \left\| v_{z,t} - \hat{h}_{z,t} \right\|^2,$$

and combining the above inner and outer iteration, the final complexity result is obtained

**Theorem 1** (Finite sum case). *For finite sum case, under assumptions 1 and 2, let $\{x_t, y_t, z_t\}$ be a sequence generated by Algorithm 1, choose $\eta_y \leq \frac{1}{2\kappa}, \eta_z \leq \frac{1}{L_{\max}\kappa}, \eta_x \leq \frac{1}{\mathcal{O}(L_{\max}^2\kappa^4)}, p = \frac{1}{\sqrt{n}},$ $b = \sqrt{n}, B = n, v \leq \frac{\epsilon^2}{\mathcal{O}(\sqrt{d_1 + d_2}\kappa^4 L_{\max}^5)}$ to let $\delta \leq \frac{\epsilon^2}{\mathcal{O}(L_{\max}^5\kappa^4)}, h = \frac{2}{r_z}\sqrt{\frac{\delta}{\rho}},$ we obtain:*

$$\mathbb{E}[\psi_{t+1} - \psi_t] \leq -\mathbb{E}[\frac{\eta_x}{2}(\|\nabla\Phi(x_t)\|^2 - 2\epsilon^2)],$$

*and total oracle cost is $\#funtion = dT(pn + b) + dn = \mathcal{O}((d_1 + d_2)\sqrt{n}\epsilon^{-2}L_{\max}^2\kappa^4)$. The detailed proof can be found in Theorem E.1 of Appendix.*

**Remark 2.** *We emphasize the computation of total function queries: each gradient estimator $\hat{\nabla}$ requires $\mathcal{O}(d_1 + d_2)$ function queries. In expectation, each iteration costs $(pB + (1 - p)b) \cdot \mathcal{O}(1)$ gradient estimators. Therefore, the total number of function queries is $\mathcal{O}((d_1 + d_2)(pB + (1 - p)b)T)$.*

**Theorem 2** (Expectation case). *For expecation case, under assumptions 1 and 2, let $\{x_t, y_t, z_t\}$ be a sequence generated by Algorithm 1, we choose $\eta_y \leq \frac{1}{2\kappa}, \eta_z \leq \frac{1}{L\kappa}, \eta_x \leq \frac{1}{\mathcal{O}(L^2\kappa^4)}, p = \frac{\epsilon}{\sigma\kappa^2},$ $b = \sigma\epsilon^{-1}\kappa^2, v \leq \frac{\epsilon^2}{\sqrt{d_1 + d_2}\kappa^4 L_{\max}^5}$ to let $\delta \leq \frac{\epsilon^2}{\mathcal{O}(L_{\max}^5\kappa^4)}, h = \frac{2}{r_z}\sqrt{\frac{\delta}{\rho}}, B \geq \mathcal{O}(L_{\max}^2\kappa^4\epsilon^{-2}\sigma^2),$ we can find the stationary point in $T = \frac{\epsilon^{-2}}{\eta_x} = \mathcal{O}(\epsilon^{-2}L_{\max}^2\kappa^4)$, and total oracle cost is $\#funtion = \mathcal{O}((d_1 + d_2)\sigma\epsilon^{-3}L_{\max}^4\kappa^8)$. The detailed proof can be found in Theorem E.2 of Appendix.*

## 4 DISCUSSION

### 4.1 IMPROVED DIMENSION DEPENDENCE COMPARED TO (AGHASI AND GHADIMI, 2024)

To simplify notation, let $d = d_1 + d_2$. The improved efficiency of our method with respect to dimensionality $d$ stems from two key factors: (1) Our single-loop structure eliminates the inner loop in (Aghasi and Ghadimi, 2024), saving $\mathcal{O}(d)$ complexity from solving inner problems. (2) We use a finite-difference approximation for Hessian-vector or Jacobian-vector products, requiring only two gradient evaluations, with $\mathcal{O}(d)$ oracle cost. This is significantly more efficient than the Gaussian smoothing method in (Aghasi and Ghadimi, 2024) Proposition 2.5, where approximating the Hessian-vector product incurs a variance of $\mathcal{O}(d^3)$ due to term $(d + 4)(d + 2)\|\nabla_{xx}^2 q\|_F^2 = \mathcal{O}(d^3\|\nabla_{xx}^2 q\|_2^2)$, leading to $\mathcal{O}(d^3)$ variance for hypergradient estimation and $\mathcal{O}(d^3)$ outer loop iterations. This is slower than our estimation method by a factor of $d^2$. Combining the two above, our complexity is $d^3$ faster than that of (Aghasi and Ghadimi, 2024).

### 4.2 DISCUSSION ON $\epsilon$ AND $d_1, d_2$ DEPENDENCY

We first illustrate that both single-level problems and min-max problems are special cases of bilevel problems. By setting $f(x, y) = f(x)$, we recover a single-level problem, and by taking $g(x, y) = -f(x, y)$, we obtain a min-max problem. This implies that the lower bounds for these problems are valid for bilevel optimization under the same assumptions.

1. Our dimension dependency, $\mathcal{O}(d_1 + d_2)$, matches the best-known results for simpler min-max problems (Wang et al., 2023; Xu et al., 2023). Moreover, our $\epsilon$ dependency aligns with the state-of-the-art complexity for first-order nonconvex-strongly convex (NC-SC) bilevel problems (Yang et al., 2023; Chu et al., 2024; Dagréou et al., 2024), establishing our dependency on $\epsilon$ and $d_1, d_2$ as the "best-known" results.

2. Since (Duchi et al., 2015) establishes a $\Omega(d)$ lower bound for single-level smooth convex problems, which are simpler than NC-SC bilevel problems (e.g., by setting $f(x, y) = f(x)$), a $\Omega(d_1)$ dependency is inevitable.

3. Our assumptions align with those used to derive the lower bound in (Dagréou et al., 2024) for NC-SC finite-sum bilevel problems, confirming that the $\mathcal{O}(\sqrt{n}\epsilon^{-2})$ dependency is inevitable in the finite-sum case .

4. The $\mathcal{O}(\epsilon^{-3})$ lower bound for single-level problems (Arjevani et al., 2023) in the expectation case is constructed under the mean-square-smooth assumption, which is slightly stronger than the typical assumption of smoothness about smoothness of $f(x, y, \xi)$ for all $\xi$. Thus, strictly speaking, $O(\epsilon^{-3})$ is not necessarily optimal in $\epsilon$ for the expectation case.

Althoud based on existing lower bounds, the dependency on $d_1$ is inevitable,however,the optimality of the $d_2$ dependency remains unclear. It would be particularly interesting if the exponent of $d_2$ could be reduced to less than 1 for lower level strongly convex optimization, as this would imply a theoretical speedup of zeroth-order algorithms compared to first-order algorithms (assuming the gradient computation cost is $\mathcal{O}(d_1 + d_2)$).

## 5 EXPERIMENTS

### 5.1 HYPER-REPRESENTATION

We verify our algorithms on hyper-representation (HR) with linear and two-layer network embedding models using synthetic data, as also discussed in (Sow et al., 2021). The hyper-representation (HR) problem (Franceschi et al., 2018; Grazzi et al., 2020a) aims to find a regression model through a two-phased optimization process. The inner level determines the optimal parameters $w$ for the linear regressor, while the outer level seeks the optimal parameters $\lambda$ for the embedding model (i.e., representation). Mathematically, this problem can be framed as the following bilevel optimization:

$$\min_{\lambda \in \mathbb{R}^{d_1}} f(\lambda) = \frac{1}{2n_1} \|T(X_1; \lambda)w^* - Y_1\|^2, \text{s.t.} w^* = \arg \min_{w \in \mathbb{R}^{d_2}} \frac{1}{2n_2} \|T(X_2; \lambda)w - Y_2\|^2 + \frac{\gamma}{2}\|w\|^2, \tag{18}$$

where $X_2 \in \mathbb{R}^{n_2 \times m}$ and $X_1 \in \mathbb{R}^{n_1 \times m}$ represent the synthesized training and validation data matrices, and $Y_2 \in \mathbb{R}^{n_2}$, $Y_1 \in \mathbb{R}^{n_1}$ are their corresponding response vectors. For the shallow HR scenario, the embedding function $T(\cdot; \lambda)$ is a linear embedding model . The data matrices $X_1, X_2$ and labels $Y_1, Y_2$ are generated using the same methodology as described in (Grazzi et al., 2020a).

We first compare the effect of different $h$ and $v$ on the zeroth-order estimator. As shown in Figure 1(a) and 1(b), we observe that the zeroth-order estimator is more accurate when $h$ and $v$ are set to 0.05 and 0.005. Second, we evaluate our proposed VRZSBO algorithm against baseline bilevel optimizers AID-FP, AID-CG, ESJ, and HOZOG (see Appendix B for details on the baselines and hyperparameters). Figures 2(a) and 2(b) display performance comparisons on linear models with dimensions 128 and 256. In both cases, VRZSBO performs comparably to or even better than the first-order methods.

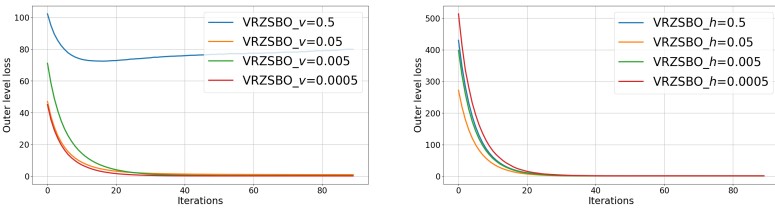

(a) Compare the effect of different $v$ on zeroth-order estimator

(b) Compare the effect of different $h$ on zeroth-order estimator

Figure 1: Zeroth-order tuning parameters

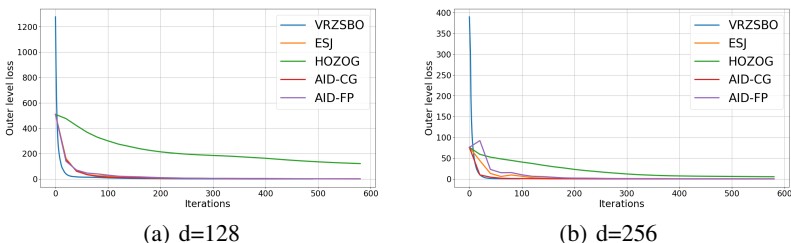

(a) d=128

(b) d=256

Figure 2: Comparison of algorithms on hyper-representation

## 5.2 HYPER-CLEANING

We compare the performance of our VRZSBO algorithm with several bilevel optimization methods, including $F^2SA$ (Kwon et al., 2023), SPABA (Chu et al., 2024), AID-FP (Grazzi et al., 2020b), ZDSBA (Aghasi and Ghadimi, 2024), and BSA (Ghadimi and Wang, 2018), in terms of runtime. The comparison is conducted on a low-dimensional data hyper-cleaning problem (Yang et al., 2023) using a linear classifier on the MNIST dataset, which is formulated as follows.

$$\min_{\lambda} L_{\nu}(\lambda, w^*) = \frac{1}{|S_{\nu}|} \sum_{(x_i, y_i) \in S_{\nu}} L_{CE}((w^*)^T x_i, y_i) \tag{19}$$

$$\text{s.t.} \quad w^* = \arg\min_{w} L(\lambda, w) := \frac{1}{|S_{\tau}|} \sum_{(x_i, y_i) \in S_{\tau}} \sigma(\lambda_i) L_{CE}(w^T x_i, y_i) + C\|w\|^2, \tag{20}$$

Here, $L_{CE}$ represents the cross-entropy loss, $S_V$ and $S_T$ denote the validation and training datasets, with sizes set to $20,000$ and $5,000$, respectively. The parameters $\lambda = \{\lambda_i\}_{i \in S_{\tau}}$ and $C$ are regularization terms, and $\sigma(\cdot)$ denotes the sigmoid function. In our experiments, we achieve superior runtime performance compared to the current state-of-the-art zeroth-order bilevel algorithm ZDSBA, demonstrating the efficiency of our method. Although our algorithm does not yet achieve the same performance as the best first-order methods, it retains the key advantage of being fully zeroth-order, making it suitable for scenarios where gradient information is unavailable.

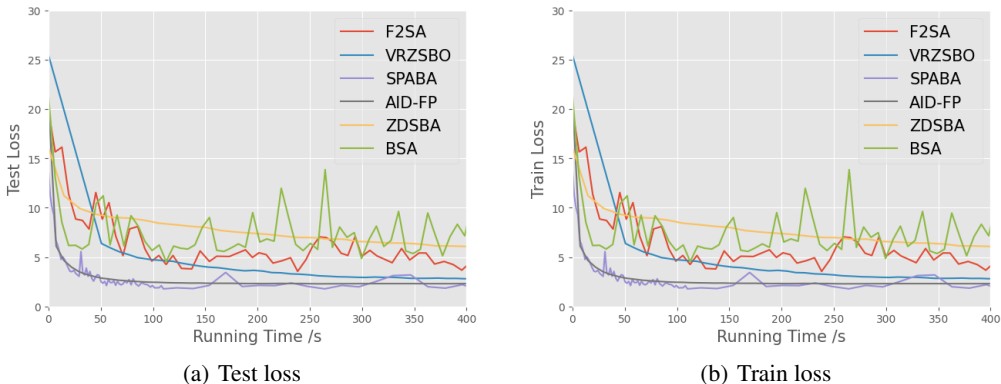

(a) Test loss        (b) Train loss

Figure 3: Comparison of algorithms on hyper-cleaning

## 6 CONCLUSION

In this paper, we propose VRZSBO, a zeroth-order algorithm for nonconvex-strongly convex bilevel optimization. Our algorithm improves the best-known complexity from $\mathcal{O}(\epsilon^{-3}(d_1 + d_2)^4)$ (Aghasi and Ghadimi, 2024) to $\mathcal{O}(\epsilon^{-3}(d_1 + d_2))$, achieving Best-known dependence on $\epsilon$ and dimension. We also analyze the finite sum case, yielding a complexity of $\mathcal{O}(\epsilon^{-2}\sqrt{n}(d_1 + d_2))$. Experiments validate the effectiveness of our algorithm. Future work may investigate the use of rand zeroth-order estimators for similar complexity and the extension of zeroth-order methods to additional problems, such as NC-PL bilevel optimization (Kwon et al., 2024; Chen et al., 2024).

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

CONTENTS

## A FURTHER DETAILS ON HYPERGRADIENT ESTIMATION

1. Neumann series (Yang et al., 2021; Khanduri et al., 2021): This method is based on the well-known Neumann series, which approximates the inverse of a matrix as:

$$\eta \sum_{i=0}^{\infty} (I - \eta \nabla_{yy}^2 g(x, y^*(x)))^i = [\nabla_{yy}^2 g(x, y(x))]^{-1}.$$

In practice, since we do not let $i \to \infty$, it can be shown that the error term $\left\| \eta \sum_{i=0}^{K} (I - \eta \nabla_{yy}^2 g(x, y^*(x)))^i - [\nabla_{yy}^2 g(x, y(x))]^{-1} \right\|$ decreases exponentially (Ghadimi and Wang, 2018) with $K$. Thus, $\nabla \Phi(x)$ can be approximated by:

$$\bar{\nabla} \Phi(x) = \nabla_x f(x, y) - \frac{K}{L} \nabla_{xy}^2 g(x, y) \sum_{k=1}^{K} \left( I - \frac{\nabla_{yy}^2 g(x, y)}{L_g} \right)^k \nabla_y f(x, y).$$

The main limitation of this approach is that it requires many matrix computations, which are expensive in practice.

2. Quadratic auxiliary function (Dagr'eou et al., 2022; Dagréou et al., 2024; Yang et al., 2023; Chu et al., 2024): let $z^* = [\nabla_{yy}^2 g(x, y^*(x))]^{-1} \nabla_y f(x, y^*(x))$. We can rewrite this as $\nabla_{yy}^2 g(x, y^*(x))z = \nabla_y f(x, y^*(x))$. Since $\nabla_{yy}^2 g(x, y^*(x))$ is invertible, this linear system has a unique solution $z^*$, which is also the minimizer of the following quadratic function:

$$R(x, y, z) := \frac{1}{2} z^T \nabla_{yy}^2 g(x, y^*(x))z - z^T \nabla_y f(x, y^*(x)). \tag{21}$$

By performing gradient descent on $R(x, y, z)$, we obtain an approximate solution $z_k$, and we approximate $\nabla \Phi(x)$ as:

$$\bar{\nabla} \Phi(x) = \nabla_x f(x, y) + \nabla_{xy}^2 g(x, y)z_k.$$

Our paper also adopts this approach, which was first introduced in (Arbel and Mairal, 2022; Dagr'eou et al., 2022). (Dagr'eou et al., 2022) achieved a complexity of $\mathcal{O}(n^{\frac{2}{3}} \epsilon^{-2})$ in the finite-sum setting using Hessian-vector product oracles and a third-order smoothness assumption. Later, (Dagréou et al., 2024) improved this to the optimal $\mathcal{O}(n^{\frac{1}{2}} \epsilon^{-2})$. (Yang et al., 2023) extended the results to the expectation setting, achieving the optimal $\mathcal{O}(\epsilon^{-3})$ rate. Incorporating the PAGE method (Li et al., 2020), (Chu et al., 2024) achieved optimal rates in finite sum and near-optimal rates expectation case.

3. Penalty method((Ye et al., 2022; Kwon et al., 2023; 2024; Chen et al., 2024)): This kind of method based on an observation that the gradient of a value function $\min_{y \in \mathbb{R}^{d_2}} g(x, y)$ can be easily expressed as :

$$\nabla g(x, y^*(x)) = \nabla_x g(x, y^*(x)) + \nabla y^*(x) \underbrace{\nabla_y g(x, y^*(x))}_{=0} = \nabla_x g(x, y^*(x)), \tag{22}$$

this expression gives us the most significant insight: the gradient of the value function is independent of the Jacobian matrix $\nabla y^*(x)$. To utilize the observation mentioned above, let us first define the penalty function:$\mathcal{L}_\lambda(x, y) := f(x, y) + \lambda(g(x, y) - g(x, y^*(x)))$, and auxiliary function:

$$\mathcal{L}_\lambda^*(x) := f(x, y_\lambda^*(x)) + \lambda(g(x, y_\lambda^*(x)) - g(x, y^*(x))),$$

where $y_\lambda^*(x) = \arg\min_{y \in \mathbb{R}^{d_2}} \mathcal{L}_\lambda(x, y)$. It has been shown (Kwon et al., 2023) that the gradient $\nabla \mathcal{L}_\lambda^*(x)$ serves as a good approximation of $\nabla \Phi(x)$, with an error bound given by $\|\nabla \mathcal{L}_\lambda^*(x) - \nabla \Phi(x)\| \leq \mathcal{O}\left(\frac{1}{\lambda}\right)$. More interestingly, utilizing the observation from (22), the gradient $\nabla \mathcal{L}_\lambda^*(x)$ takes on a simplified form:

$$\nabla \mathcal{L}_\lambda^*(x) = \nabla_x f(x, y_\lambda^*(x)) + \lambda (\nabla_x g(x, y_\lambda^*(x)) - \nabla_x g(x, y^*(x))).$$

The key advantage is that, unlike the gradient of $\Phi(x)$ in (1), the gradient of $\mathcal{L}_\lambda(x)$ involves only first-order terms, avoiding the need to compute second-order terms. The main challenge, however, is that the approximation error of $y^*(x)$ and $y_\lambda^*(x)$, used to estimate $\nabla\mathcal{L}_\lambda(x)$, is multiplied by $\lambda$. Thus, obtaining accurate estimators for $y^*(x)$ and $y_\lambda^*(x)$ is crucial. To our best knowledge, (Kwon et al., 2023) first proved this framework in the stochastic case, but it has suboptimal complexities of $\mathcal{O}(\epsilon^{-3})$ in the deterministic case and $\mathcal{O}(\epsilon^{-5})$ in the expectation case. (Chen et al., 2023) later improved the deterministic complexity to a near-optimal $\tilde{\mathcal{O}}(\epsilon^{-2})$, (Kwon et al., 2024) later reduced the complexity of expectation case to $\tilde{\mathcal{O}}(\epsilon^{-4})$, still leaving a gap of $\mathcal{O}(\epsilon^{-1})$ from the optimal rate of single-level problem(i.e,$\epsilon^{-3}$).

Consequently, stochastic bilevel optimization with strongly-convex lower-level problems is now as efficient as single-level optimization, prompting the exploration of more challenging scenarios, such as zeroth-order optimization (where first-order oracles are inaccessible) and cases lacking strong convexity at the lower level (Huang, 2024; Chen et al., 2024; Kwon et al., 2024).

## B DETAILS FOR EXPERIMENT

### B.1 HYPER-REPRESENTATION

We first take the following introduction from Sow et al. (2021):

- HOZOG (Gu et al., 2021): a hyperparameter optimization algorithm that uses evolution strategies to estimate the entire hypergradient (both the direct and indirect component). We use our own implementation for this method.

- AID-CG (Grazzi et al., 2020a): approximate implicit differentiation with conjugate gradient. We use its implementation provided at https://github.com/prolearner/hypertorch

- AID-FP (Grazzi et al., 2020a): approximate implicit differentiation with fixed-point. We experimented with its implementation at the repository https://github.com/prolearner/hypertorch

- ITD-R (REVERSE) (Franceschi et al., 2017): an iterative differentiation method that computes hypergradients using reverse mode automatic differention (RMAD). We use its implementation provided at https://github.com/prolearner/hypertorch.

**Hyperparameters setup** For VRZSBO we use $v = \{0.01, 0.005\}$, $h = \{0.01, 0.005\}$,$\eta_x = 0.00005$, $\eta_z = 0.0001$, $\eta_y = 0.001$,for HOZOG we set $\mu = 0.01$.For other method,the number of inner GD steps is fixed to$N = 20$ with the learning rate of $\alpha = 0.001$. For the outer optimizer, we use Adam with a learning rate of 0.05. The value of $\gamma$ in (18) is set to be 0.1. For all double-loop methods, we set $N = 10$ , $\alpha = 0.001$, $\beta = 0.001$, and use Adam with a learning rate of 0.01 as the outer optimizer.

### B.2 HYPER-CLEANING

For our VRZSBO,we set $v = 0.0001, h = 0.001$,$\eta_x = 0.1, \eta_y = 0.5, \eta_z = 0.5$,$B = 1000, b = 100, p = 0.5$.For ZDSBA,we set $v = 0.0001, h = 0.001$,$\eta_x = 0.001, \eta_y = 0.01, \eta_z = 0.001$.For SPABA ,we set $\eta_x = 0.1, \eta_y = 0.5, \eta_z = 0.5$,$B = 1000, b = 100, p = 0.5$.We set the number of inner-loop iterations to 20 for AID-FP,F$^2$SA and BSA,and choose 0.1 as the both inner and outer stepsize.

Table 2: Meaning of Symbols

| Symbol | Meaning |
|---|---|
| $v$ | Smoothing parameter used in gradient estimation |
| $h$ | Smoothing parameter used in estimating $Hv/Jv$ |
| $C_f$ | Lipschitz constant of $f(x, y)$ in $y$ |
| $L$ | Lipschitz constant of $f(x, y)$, $g(x, y)$, and their sample functions |
| $\mu$ | Strong convexity constant of $g(x, y)$ in $y$ |
| $r_z$ | Upper bound of $\|z_t^*\|$ |
| $\rho$ | Hessian-Lipschitz constant of $g(x, y)$ and its sample functions |
| $\delta$ | Upper bound on the error between the zeroth-order estimator and the gradient |
| $H_v(t)$ | Hessian vector product $\nabla_{yy}^2 g(x_t, y_t) z_t$ |
| $DH_v(t)$ | Finite difference that approximates the vector product $\nabla_{xy}^2 g(x_t, y_t) z_t$ |
| $\hat{DH}_v(t)$ | Zeroth-order estimator for $DH_v(t)$ |
| $DH_v(t; \xi)$ | Stochastic sample estimator for $DH_v(t)$ |
| $L_{max}$ | Largest smoothness constant |
| $\kappa$ | Largest condition number |

## C  USEFUL LEMMAS

**Lemma C.1** ((Xu et al., 2017)). *For $\rho$-Hessian Lipschitz continuous function $f(x)$, we have we have*

$$\left\| \nabla f(\mathbf{x} + \mathbf{u}) - \nabla f(\mathbf{x}) - \nabla^2 f(\mathbf{x})\mathbf{u} \right\| \leq \frac{\rho \|\mathbf{u}\|^2}{2}. \tag{23}$$

**Lemma C.2** (Jensen's inequality). *For convex function $f(x)$ we have*

$$f(\mathbb{E}[x]) \leq \mathbb{E}[f(x)], \tag{24}$$

*two extended versions of Jensen's inequality are*

$$\|\mathbb{E}[x]\| \leq \mathbb{E}[\|x\|], \text{ for } x \in \mathbb{R}^d$$

$$\left\| \sum_{i=1}^{k} a_i \right\|^2 \leq k \sum_{i=1}^{k} \|a_i\|^2, \text{ for } a_i \in \mathbb{R}^d.$$

**Lemma C.3** (Young's inequality). *For any vectors $a, b, \in \mathbb{R}^d$, and $\zeta \geq 0$, the following inequality holds:*

$$\|a\|^2 \leq (1 + \zeta)\|a - b\|^2 + \left(1 + \zeta^{-1}\right)\|b\|^2,$$

*an extended version of Young's inequality is*

$$\langle a, b \rangle \leq \frac{\|a\|^2}{2\zeta} + \frac{\zeta\|b\|^2}{2}.$$

**Lemma C.4** (variance decomposition). *For random vector $x \in \mathbb{R}^d$ and any $y \in \mathbb{R}^d$, the variance of $x$ can be decomposed as*

$$\mathbb{E}\left[\|x - \mathbb{E}[x]\|^2\right] = \mathbb{E}\left[\|x - y\|^2\right] - \mathbb{E}\left[\|\mathbb{E}[x] - y\|^2\right],$$

*which implies*

$$\mathbb{E}\left[\|x - \mathbb{E}[x]\|^2\right] \leq \mathbb{E}\left[\|x\|^2\right].$$

**Lemma C.5.** *For random variable $X, Y$, if $X$, $Y$ are independent, and $\mathbb{E}[X]$ or $\mathbb{E}[Y] = 0$, we have*

$$\mathbb{E}[\|X - Y\|^2] = \mathbb{E}[\|X\|^2] + \mathbb{E}[\|Y\|^2]. \tag{25}$$

*Proof.*

$$\mathbb{E}[\|X - Y\|^2] = \mathbb{E}[\|X\|^2 + \|Y\|^2 + 2\mathbb{E}\langle X, Y \rangle] = \mathbb{E}[\|X\|^2] + \mathbb{E}[\|Y\|^2].$$

$\square$

**Lemma C.6.** *For i.i.d. $x_1, x_2, x_3 \cdots x_n$, if $\mathbb{E}[x_i] = x, \mathbb{E}[\|x_i - x\|^2] \leq \sigma^2$, we have*

$$\mathbb{E}\left[\left\|\frac{1}{b}\sum_{i=1}^{b} x_i - x\right\|^2\right] \leq \frac{\mathbb{E}[\|x_i\|^2]}{b}. \tag{26}$$

*Proof.*

$$\mathbb{E}\left[\left\|\frac{1}{b}\sum_{i=1}^{b} x_i - x\right\|^2\right]$$

$$= \frac{1}{b^2}\mathbb{E}\left[\left\|\sum_{i=1}^{b}(x_i - x)\right\|^2\right]$$

$$= \frac{1}{b^2}\sum_{i=1}^{b}\mathbb{E}[\|x_i - x\|^2]$$

$$= \frac{1}{b}\mathbb{E}[\|x_i - x\|^2] \leq \frac{\mathbb{E}[\|x_i\|^2]}{b},$$

where the second inequality holds because $\|a + b\|^2 = \|a\|^2 + \|b\|^2 + 2\langle a, b \rangle$, and $\mathbb{E}[\langle x_i - x, x_j - x \rangle] = 0 (j \neq i)$ for iid random variable $x_i$. $\square$

**Lemma C.7** ((Li et al., 2020)). *Suppose that function $f$ is $L$-smooth and let $x_{t+1} := x_t - \eta g^t$, then for any $g^t \in \mathbb{R}^d$ and $\eta > 0$, we have*

$$f(x_{t+1}) \leq f(x_t) - \frac{\eta}{2}\|\nabla f(x_t)\|^2 - \left(\frac{1}{2\eta} - \frac{L}{2}\right)\|x_{t+1} - x_t\|^2 + \frac{\eta}{2}\|g^t - \nabla f(x_t)\|^2.$$

## D  COORD ESTIMATE AND ERROR BOUND

Table 3: some bound of estimator

| Error term | Meaning | bound |
|---|---|---|
| $\left\|\hat{\nabla}p(x) - \nabla p(x)\right\|^2$ | error between gradient and its zeroth-order estimator | lemma D.1 |
| $\left\|\hat{\nabla}p(x;\xi) - \hat{\nabla}p(x)\right\|^2$ | variance of zeroth-order estimator | lemma D.2 |
| $\left\|\hat{\nabla}p(x_1;\xi) - \hat{\nabla}p(x_2;\xi)\right\|^2$ | the Lipschitzness of sample zeroth-order gradient estimator | lemma D.3 |
| $\|DH_v(t) - H_v(t)\|^2$ | the error between finite difference and Hessian vector product | lemma D.5 |
| $\left\|D\hat{H}_v(t) - DH_v(t)\right\|^2$ | the error between $D\hat{H}_v(t;\xi)$ and its sample estimator | lemma D.6 |
| $\left\|D\hat{H}_v(t;\xi) - D\hat{H}_v(t)\right\|^2$ | the Lipschitzness of sample zeroth-order Hv estimator | lemma D.7 |

## D.1  ERROR BOUND

### D.1.1  ZEROTH-ORDER ESTIMATOR FOR GRADIENT

**Lemma D.1** (Restatement of lemma 2). *For $L$-smooth function $p(x)$, its gradient $\nabla p(x)$ and its zeroth-order estimator $\hat{\nabla} p(x)$, we have*

$$\left\| \hat{\nabla} p(x) - \nabla p(x) \right\|^2 \leq \frac{L^2}{4} d v^2. \tag{27}$$

*Proof.*

$$\left\| \hat{\nabla} p(x) - \nabla p(x) \right\|^2 = \left\| \sum_{\ell=1}^{d} \left( \frac{\partial f_v(\mathbf{x})}{\partial x_\ell} - \frac{\partial p(\mathbf{x})}{\partial x_\ell} \right) \mathbf{e}_\ell \right\|^2$$

$$= \sum_{\ell=1}^{d} \left\| \frac{\partial f_v(\mathbf{x})}{\partial x_\ell} - \frac{\partial p(\mathbf{x})}{\partial x_\ell} \right\|^2 \leq \frac{L^2}{4} d v^2.$$

in second inequality we use $\left\| \frac{\partial f_v(\mathbf{x})}{\partial x_\ell} - \frac{\partial p(\mathbf{x})}{\partial x_\ell} \right\| = \left\| \frac{f(\mathbf{x}+v\mathbf{e}_\ell)-f(\mathbf{x})-\langle \nabla p(x), v e_\ell \rangle}{v} \right\| \leq \frac{vL}{2}$  ☐

**Corollary 2.** *Choose suffciently small zeroth-oreder smooth parameter $v$, the gradient estimation error can be bounded by a constant $\delta$:*

$$\left\| \hat{\nabla} f(c) - \nabla f(c) \right\| \leq \delta. \tag{28}$$

*and*

$$\left\| \hat{\nabla} g(c) - \nabla g(c) \right\| \leq \delta. \tag{29}$$

where $\delta = \frac{L\sqrt{d_1+d_2}v}{2}$. In the following lemma, we disscuss the error between zeroth-order estimator $\hat{\nabla} p(x)$ and stochastic zeroth-order estimator $\hat{\nabla} p(x;\xi)$,

**Lemma D.2.** *Under assumptions 1 and 2, we have*

$$\mathbb{E}[\left\| \hat{\nabla} p(x;\xi) - \hat{\nabla} p(x) \right\|^2] \leq 6\delta^2 + 3\sigma^2. \tag{30}$$

*Proof.*

$$\mathbb{E}\left[ \left\| \hat{\nabla} p(x;\xi) - \hat{\nabla} p(x) \right\|^2 \right]$$

$$\leq 3 \left\| \hat{\nabla} p(x;\xi) - \nabla p(x;\xi) \right\|^2 + 3 \left\| \hat{\nabla} p(x) - \nabla p(x) \right\|^2 + 3\mathbb{E}[\|\nabla p(x) - \nabla p(x;\xi)\|^2]$$

$$\leq 6\delta^2 + 3\sigma^2,$$

where the last inequality holds because $\left\| \hat{\nabla} p(x;\xi) - \nabla p(x;\xi) \right\|^2$ and $\left\| \hat{\nabla} p(x) - \nabla p(x) \right\|^2$ are bounded by $\delta^2$, and $\mathbb{E}[\|\nabla p(x) - \nabla p(x;\xi)\|^2] \leq \sigma^2$.  ☐

In the following lemma, we disscuss the Lipschitzness of sample zeroth-order gradient estimator.

**Lemma D.3.** *under assumptions 1 and 2, we have*

$$\left\| \hat{\nabla} p(x_1;\xi) - \hat{\nabla} p(x_2;\xi) \right\|^2 \leq 3L^2 \|x_1 - x_2\|^2 + 6\delta^2. \tag{31}$$

*Proof.*

$$\left\| \hat{\nabla} p(x_1;\xi) - \hat{\nabla} p(x_2;\xi) \right\|^2$$

$$\leq 3 \|\nabla p(x_1;\xi) - \nabla p(x_2;\xi)\|^2 + \left\| \nabla p(x_1;\xi) - \hat{\nabla} p(x_1;\xi) \right\|^2 + \left\| \nabla p(x_2;\xi) - \hat{\nabla} p(x_2;\xi) \right\|^2)$$

$$\leq 3L^2 \|x_1 - x_2\|^2 + 6\delta^2,$$

where the last inequality holds due to D.1 and assumption 1 .  ☐

**Corollary 3.** *since $f(x, y)$ and $g(x, y)$ satisfy assumptions 1 and 2, the conclusion of above lemma D.2, D.3 holds for $g(x, y)$ and $f(x, y)$.*

### D.1.2 ZEROTH-ORDER ESTIMATOR FOR HESSIAN-VECTOR PRODUCT

**Remark 3.** *we only prove the conclusion about Hessian vector product, the proof and conclusions of Jacobian vector product are the same with same as the former .*

In the following lemma, we disscuss the error between finite difference(defined in (7)) and its zeroth-order estimator(defined in (9)).

**Lemma D.4.** *Under assumption 1, we have*

$$\left\| D\hat{H}_v(t) - DH_v(t) \right\| \leq \frac{2\delta}{h}, \tag{32}$$

*Proof.*

$$\left\| D\hat{H}_v(t) - DH_v(t) \right\|$$
$$\leq \left\| \frac{\hat{\nabla}_y g(x_t, y_t + hz_t) - \nabla_y g(x_t, y_t + hz_t)}{h} \right\| + \left\| \frac{\hat{\nabla}_y g(x_t, y_t) - \nabla_y g(x_t, y_t)}{h} \right\|$$
$$\leq \frac{2\delta}{h}.$$

$\square$

In the following lemma, we disscuss the error between finite difference(defined in (7)) and Hessian vector product(defined in (6)).

**Lemma D.5.** *Under assumption 1, we have*

$$\|DH_v(t) - H_v(t)\| \leq \frac{\rho h r_z^2}{2} \tag{33}$$

*Proof.*

$$\|DH_v(t) - H_v(t)\| = \left\| \frac{\hat{\nabla}_y g(x, y + hz_t) - \hat{\nabla}_y g(x, y)}{h} - \nabla_{yy}^2 g(x, y) z_t \right\|$$
$$= \frac{1}{h} \left\| \hat{\nabla}_y g(x, y + hz_t) - \hat{\nabla}_y g(x, y) - \nabla_{yy}^2 g(x, y) h z_t \right\|$$
$$\leq \frac{1}{h} \frac{\rho \|hz_t\|^2}{2}$$
$$\leq \frac{\rho h r_z^2}{2},$$

where the third inequality holds due to lemma C.1. $\square$

In the following lemma, we disscuss the error between finite difference(defined in (9)) and Hessian vector product(defined in (6)).

**Lemma D.6.** *Under assumption 1, choose $h = \frac{2}{r_z}\sqrt{\frac{\delta}{\rho}}$, we have*

$$\left\| D\hat{H}_v(t) - H_v(t) \right\|^2 \leq 4r_z^2 \rho \delta. \tag{34}$$

*Proof.*

$$\left\| D\hat{H}_v(t) - H_v(t) \right\|^2$$

$$\leq 2 \left\| D\hat{H}_v(t) - DH_v(t) \right\|^2 + 2 \left\| DH_v(t) - H_v(t) \right\|^2$$

$$\leq 2\left(\frac{4\delta^2}{h^2} + \frac{\rho^2 h^2 r_z^4}{4}\right)$$

$$\leq 4r_z^2 \rho\delta,$$

where the last inequality holds because we choose $h = \frac{2}{r_z}\sqrt{\frac{\delta}{\rho}}$. $\qquad\square$

In the following lemma, we discuss the error between $D\hat{H}_v(t;\xi)$(defined in (9)) and its sample estimator(defined in (11)).

**Lemma D.7** (variance of sample zeroth-order estimator)**.** *Under assumptions 1 and 2, for our zeroth-order estimator, we have*

$$\left\| D\hat{H}_v(t;\xi) - D\hat{H}_v(t) \right\|^2 \leq 3r_z^2\sigma^2 + 24r_z^2\rho\delta.$$

*Proof.*

$$\left\| D\hat{H}_v(t;\xi) - D\hat{H}_v(t) \right\|^2$$

$$\leq 3 \left\| H_v(t;\xi) - H_v(t) \right\|^2 + 3 \left\| D\hat{H}_v(t;\xi) - H_v(t;\xi) \right\|^2 + 3 \left\| H_v(t) - D\hat{H}_v(t) \right\|^2$$

$$\leq 3r_z^2\sigma^2 + 12r_z^2\rho\delta + 12r_z^2\rho\delta.$$

where the last inequality holds due to lemma D.6, and $\|H_v(t;\xi) - H_v(t)\|^2 = \left\| \left(\nabla^2_{yy}g(x_t,y_t) - \nabla^2_{yy}g(x_t,y_t;\xi)\right)z_t \right\|^2 \leq \sigma^2 \|z_t\|^2$. $\qquad\square$

In the following lemma, we disscuss the Lipschitzness of sample zeroth-order Hv estimator.

**Lemma D.8** (Lipschitzness of the zeroth order estimator for Hv)**.** *Under assumptions 1 and 2, we have*

$$\left\| D\hat{H}_v(x_{t+1};\xi) - D\hat{H}_v(x_t;\xi) \right\|^2 \leq 8r_z^2 \|c_{t+1} - c_t\|^2 + 8L^2 \|z_{t+1} - z_t\|^2 + 16r_z^2\rho\delta. \qquad (35)$$

*Proof.*

$$\left\| D\hat{H}_v(x_{t+1};\xi) - D\hat{H}_v(x_t;\xi) \right\|^2$$

$$\leq 4(\|H(x_{t+1};\xi) - H(x_t;\xi)\|^2 + \left\| D\hat{H}_v(x_t;\xi) - H(x_t;\xi) \right\|^2 + \left\| D\hat{H}_v(x_{t+1};\xi) - H(x_{t+1};\xi) \right\|^2)$$

$$\leq 8 \left\| (\nabla^2_{yy}g(x_{t+1},y_{t+1};\xi) - \nabla^2_{yy}g(x_t,y_t;\xi))z_t \right\|^2 + 8 \left\| \nabla^2_{yy}g(x_t,y_t;\xi)(z_{t+1} - z_t) \right\|^2 + 16r_z^2\rho\delta$$

$$\leq 8r_z^2 \|c_{t+1} - c_t\|^2 + 8L^2 \|z_{t+1} - z_t\|^2 + 16r_z^2\rho\delta,$$

where the second inequality holds due to the conclusion of lemma D.6. $\qquad\square$

## E  CONVERGENCE ANALYSIS

Table 4: lemma used in converge analysis

| Meaning | lemma |
|---|---|
| inexact gradient descent of $z$ | lemma E.6 |
| variance descent $z$ (finite sum case) | lemma E.7 |
| variance descent $z$ (Expectation case) | lemma E.8 |
| inexact gradient descent of $y$ | lemma E.12 |
| variance descent of $y$ (finite sum case) | lemma E.13 |
| variance descent of $y$ (Expectation case) | lemma E.14 |
| inexact gradient descent of $\Phi(x)$ | lemma E.18 |
| variance descent of $x$ (finite sum case) | lemma E.19 |
| variance descent of $x$ (Expectation case) | lemma E.20 |
| converge analysis of $\Phi(x)$ (finite sum case) | Therorem E.1 |
| converge analysis of $\Phi(x)$ (Expectation case) | Therorem E.2 |

### E.1  BASCI PROPERTY ABOUT BILEVEL PROBLEM

**Lemma E.1** (bound of $\|z_t^*\|$). *Under assumption 1, for any $z_t^* = [\nabla_{yy}^2 g(x_t, y_t)]^{-1}\nabla_y f(x_t, y_t)$, we have*

$$\|z_t^*\| \leq r_z. \tag{36}$$

*where $r_z := \frac{C_f}{\mu}$*

*Proof.*

$$\|z_t^*\| = \left\|[\nabla_{yy}^2 g(x_t, y_t)]^{-1}\nabla_y f(x_t, y_t)\right\| \leq \left\|[\nabla_{yy}^2 g(x_t, y_t)]^{-1}\right\| \|\nabla_y f(x_t, y_t)\| \leq \frac{C_f}{\mu}.$$

$\square$

**Lemma E.2** (Lipschitzness of $z^*$). *Under assumption 1 we have*

$$\left\|z_{t+1}^* - z_t^*\right\| \leq L_{Z^*}\left\|c_{t+1} - c_t\right\|. \tag{37}$$

*where $L_{Z^*} := (\frac{L}{\mu} + \frac{C_f\rho}{\mu^2}) = \mathcal{O}(\kappa^2)$*

*Proof.*

$$
\begin{aligned}
\left\|z_{t+1}^* - z_t^*\right\| &= \left\|[\nabla_{yy}^2 g(x_{t+1}, y_{t+1})]^{-1}\nabla_y f(x_{t+1}, y_{t+1}) - [\nabla_{yy}^2 g(x_t, y_t)]^{-1}\nabla_y f(x_t, y_t)\right\| \\
&\leq \left\|[\nabla_{yy}^2 g(x_{t+1}, y_{t+1})]^{-1}\right\| \|\nabla_y f(x_{t+1}, y_{t+1}) - \nabla_y f(x_t, y_t)\| \\
&\quad + \|\nabla_y f(x_t, y_t)\| \left\|[\nabla_{yy}^2 g(x_{t+1}, y_{t+1})]^{-1} - [\nabla_{yy}^2 g(x_t, y_t)]^{-1}\right\| \\
&= \left\|[\nabla_{yy}^2 g(x_{t+1}, y_{t+1})]^{-1}\right\| \|\nabla_y f(x_{t+1}, y_{t+1}) - \nabla_y f(x_t, y_t)\| \\
&\quad + \|\nabla_y f(x_t, y_t)\|^2 \left\|\nabla_{yy}^2 g(x_{t+1}, y_{t+1})(\nabla_{yy}^2 g(x_{t+1}, y_{t+1}) - \nabla_{yy}^2 g(x_t, y_t))\nabla_{yy}^2 g(x_t, y_t)\right\| \\
&\leq (\frac{L}{\mu} + \frac{C_f\rho}{\mu^2}) \left\|c_{t+1} - c_t\right\|.
\end{aligned}
$$

$\square$

### E.2 DESCENT PROPERTY IN $z$

In the following lemma, we disscuss the error bewteen $h_{z,t}$ and its zero-order estimator.

**Lemma E.3.** *Under assumptions 1 and 2, we have*

$$\left\| h_{z,t} - \hat{h}_{z,t} \right\|^2 \leq 2\delta^2 + 8r_z^2\rho\delta \tag{38}$$

*Proof.* use lemma D.6, we obtain

$$\left\| h_{z,t} - \hat{h}_{z,t} \right\|^2 \leq 2\left\| D\hat{H}_v(t) - H_v(t) \right\|^2 + 2\left\| \hat{\nabla}_y f(x_t, y_t) - \nabla_y f(x_t, y_t) \right\|^2 \leq 2\delta^2 + 8r_z^2\rho\delta$$

$\square$

**Lemma E.4.** *Under assumptions 1 and 2 we have*

$$\left\| \left( D\hat{H}_v(t;\xi) - \hat{\nabla}_y f(x_t, y_t;\zeta) \right) - \hat{h}_{z,t} \right\|^2 \leq 6(1 + r_z^2)\sigma^2 + 48r_z^2\rho\delta + 12\delta^2. \tag{39}$$

*Proof.*

$$\left\| \left( D\hat{H}_v(t;\xi) - \hat{\nabla}_y f(x_t, y_t;\zeta) \right) - \hat{h}_{z,t} \right\|^2$$

$$\leq 2\left\| D\hat{H}_v(t;\xi) - D\hat{H}_v(t) \right\|^2 + 2\left\| \hat{\nabla}_y f(x_t, y_t;\zeta) - \hat{\nabla}_y f(x_t, y_t) \right\|^2$$

$$\leq 2(3r_z^2\sigma^2 + 24r_z^2\rho\delta + 6\delta^2 + 3\sigma^2)$$

$$= 6(1 + r_z^2)\sigma^2 + 48r_z^2\rho\delta + 12\delta^2.$$

where the second inequality holds due to the conclusions of lemmas D.7 and D.2, which are used to upper bound and to upperbound $\left\| D\hat{H}_v(t;\xi) - D\hat{H}_v(t) \right\|^2$ and $\left\| \hat{\nabla}_y f(x_t, y_t;\zeta) - \hat{\nabla}_y f(x_t, y_t) \right\|^2$.

$\square$

**Lemma E.5** ((Necoara et al., 2019)). *The projected gradient descent update rule of convex function* $f$ *is given as follows:*

$$x_{k+1} = [x_k - \alpha_k \nabla f(x_k)]_X$$

*where $X$ is a closed convex set, $\alpha_k$ is the step size, and $[\cdot]_X$ denotes the projection onto $X$. We have the following property:*

$$\langle x_{k+1} - x_k + \alpha_k \nabla f(x_k), x - x_{k+1} \rangle \geq 0 \quad \forall x \in X. \tag{40}$$

In the following lemma, we disscuss the inexact descent property of $z_t$.

**Lemma E.6** (Restatement of lemma 4). *Under assumption 1, let $\{z_t\}$ be a sequence generated by Algorithm 1,we have*

$$\left\| z_{t+1} - z_{t+1}^* \right\|^2 \leq (1 - \frac{\eta_z\mu}{6})\left\| z_t - z_t^* \right\|^2 + \frac{4\eta_z}{\mu}(\left\| v_{z,t} - \hat{h}_{z,t} \right\|^2 + \left\| h_{z,t} - \hat{h}_{z,t} \right\|^2) - (1 - 2L\eta_z)\left\| z_{t+1} - z_t \right\|^2$$

$$+ \frac{4}{\eta_z\mu}l_{Z^*}^2 \left\| c_{t+1} - c_t \right\|^2.$$

*Proof.* The following proof is inspired by (Necoara et al., 2019) Theorem 11 . note that $\nabla_{zz}^2 R(x,y,z) = \nabla_{yy}^2 g(x,y)$, so $R(x,y,z)$ is $L$-smooth and $\mu$-strongly convex in $z$.

$$\|z_{t+1} - z_t^*\|^2$$

$$= \|z_{t+1} - z_t\|^2 + 2\langle z_{t+1} - z_t, z_t - z_t^*\rangle + \|z_t - z_t^*\|^2$$

$$= -\|z_{t+1} - z_t\|^2 + 2\langle z_{t+1} - z_t, z_{t+1} - z_t^*\rangle + \|z_t - z_t^*\|^2$$

$$\overset{a}{\le} -\|z_{t+1} - z_t\|^2 + 2\eta_z\left(\langle\nabla_z R(c_t, z_t), z_t^* - z_{k+1}\rangle + \left\langle \hat{h}_t^z - \nabla_z R(c_t, z_t), z_t^* - z_{t+1}\right\rangle\right)$$

$$\quad + \|z_t - z_t^*\|^2 \pm l_{R,z}\eta_z\|z_{t+1} - z_t\|^2$$

$$\overset{b}{\le} -\|z_{t+1} - z_t\|^2 + \eta_z\gamma_2\|z_t^* - z_{t+1}\|^2 + \frac{\eta_z}{\gamma_2}\left\|\hat{h}_t^z - \nabla_z R(c_t, z_t)\right\|^2 + \|z_t - z_t^*\|^2$$

$$\quad - 2\eta_z\left(\langle\nabla_z R(c_t, z_t), z_t - z_t^*\rangle + \langle\nabla_z R(c_t, z_t), z_{t+1} - z_t\rangle\right) \pm l_{R,z}\eta_z\|z_{t+1} - z_t\|^2$$

$$\overset{c}{\le} \eta_z\gamma_2\|z_t^* - z_{t+1}\|^2 + \frac{\eta_z}{\gamma_2}\left\|\hat{h}_t^z - \nabla_z R(c_t, z_t)\right\|^2 - 2\eta_z\left(R(c_t, z_{t+1}) - R(c_t, z_t^*)\right)$$

$$\quad + (l_{R,z}\eta_z - 1)\|z_{t+1} - z_t\|^2 + \|z_t - z_t^*\|^2$$

$$\overset{d}{\le} \|z_t - z_t^*\|^2 + (\eta_z\gamma_2 - \eta_z\mu)\|z_{t+1} - z_t^*\|^2 + \frac{\eta_z}{\gamma_2}\left\|\hat{h}_t^z - \nabla_z R(c_t, z_t)\right\|^2 + (l_{R,z}\eta_z - 1)\|z_{t+1} - z_t\|^2,$$

where step (a) holds because $z_t^* \in \mathbb{B}(r_z)$. Therefore, we can use lemma D.5, step (b) holds due to Young's inequality, step (c) holds due to the Lipschitzness of $R(x,y,z)$ in $z$, step (d) holds due to the strong convexity of $R(x,y,z)$ in $z$ ($(R(c_t, z_{t+1}) - R(c_t, z_t^*)) \ge \frac{\|z_{t+1}-z_t^*\|^2}{2}$). Now, let $\gamma_2 = \frac{\mu}{2}$, we obtain

$$(1 + \frac{\eta_z\mu}{2})\|z_{t+1} - z_t^*\|^2 \tag{41}$$

$$\le \|z_t - z_t^*\|^2 + \frac{2\eta_z}{\mu}\|v_{z,t} - \nabla_z R(c_t, z_t)\|^2 - (1 - L\eta_z)\|z_{t+1} - z_t\|^2, \tag{42}$$

after that,

$$\left\|z_{t+1} - z_{t+1}^*\right\|^2$$

$$= \left\|z_{t+1} \pm z_t^* - z_{t+1}^*\right\|^2$$

$$\overset{a}{\le} (1 + \gamma_3)\|z_{t+1} - z_t^*\|^2 + (1 + \frac{1}{\gamma_3})\left\|z_t^* - z_{t+1}^*\right\|^2$$

$$\le (1 + \gamma_3)\|z_{t+1} - z_t^*\|^2 + (1 + \frac{1}{\gamma_3})l_{Z^*}^2\|c_{t+1} - c_t\|^2,$$

where step (a) holds due to Young's inequality. Combine with (42), we have

$$\left\|z_{t+1} - z_{t+1}^*\right\|^2$$

$$\le \frac{(1+\gamma_3)}{(1+\frac{\eta_z\mu}{2})}\left(\|z_t - z_t^*\|^2 + \frac{2\eta_z}{\mu}\|v_{z,t} - \nabla_z R(c_t, z_t)\|^2 - (1 - L\eta_z)\|z_t - z_t\|^2\right) + \frac{1}{\gamma_3}l_{Z^*}\|c_{t+1} - c_t\|^2$$

$$\overset{a}{\le} (1 - \frac{\eta_z\mu}{6})\left(\|z_t - z_t^*\|^2 + \frac{2\eta_z}{\mu}\|v_{z,t} - \nabla_z R(c_t, z_t)\|^2 - (1 - L\eta_z)\|z_{t+1} - z_t\|^2\right) + \frac{4}{\eta_z\mu}l_{Z^*}^2\|c_{t+1} - c_t\|^2$$

$$\overset{b}{\le} (1 - \frac{\eta_z\mu}{6})\|z_t - z_t^*\|^2 + \frac{4\eta_z}{\mu}\left(\left\|v_{z,t} - \hat{h}_{z,t}\right\|^2 + \left\|h_{z,t} - \hat{h}_{z,t}\right\|^2\right) - (1 - 2L\eta_z)\|z_{t+1} - z_t\|^2 + \frac{4}{\eta_z\mu}l_{Z^*}^2\|c_{t+1} - c_t\|^2,$$

where step (a) holds due because we choose $\gamma_3 \le \frac{\eta_z\mu}{4}$, and let $\eta_z\mu \le 1$, such that $\frac{(1+\gamma_3)}{(1+\frac{\eta_z\mu}{2})} = 1 - \frac{\frac{\eta\mu}{2}-\gamma_3}{(1+\frac{\eta_z\mu}{2})} \le 1 - \frac{\frac{\eta\mu}{4}}{(1+\frac{\eta_z\mu}{2})} \le 1 - \frac{\eta_z\mu}{6}$,

step(b) holds because $(1-\frac{\eta_z\mu}{6})(1-2L\eta_z) \ge 1-\frac{\eta_z\mu}{6}-\eta_z L \ge 1-2L\eta_z$, note that $\nabla_z R(c_t, z_t) = h_{z,t}$, and $\|v_{z,t} - h_{z,t}\|^2 \le 2(\left\|v_{z,t} - \hat{h}_{z,t}\right\|^2 + \left\|h_{z,t} - \hat{h}_{z,t}\right\|^2)$, hence, we have finished proof. $\qquad\square$

In the following lemma, we disscuss the variance descent property of variance term in finite sum case.

**Lemma E.7** (Restatement of lemma 6). *Under assumptions 1 and 2, we have*

$$\mathbb{E}\left\|v_{z,t+1} - \hat{h}_{z,t+1}\right\|^2$$

$$\leq (1-p)\mathbb{E}[\left\|v_{z,t} - \hat{h}_{z,t}\right\|^2] + \frac{2(1-p)}{b}\left((8r_z^2 + 3L^2)\left\|c_{t+1} - c_t\right\|^2 + 8L^2\left\|z_{t+1} - z_t\right\|^2 + 16r_z^2\rho\delta + 6\delta^2\right).$$

*Proof.* recall that

$$v_{z,t+1} := \begin{cases} \frac{1}{B}\sum_{i=1}^{B} D\hat{H}_v(t+1;\xi_i) - \hat{\nabla}_y f(x_{t+1}, y_{t+1};\zeta_i) & \text{with probability } p, \\ v_{z,t} + \frac{1}{b}\sum_{i=1}^{b}\sum D\hat{H}_v(t+1;\xi_i) - D\hat{H}_v(t;\xi_i) \\ -\frac{1}{b}\sum_{i=1}^{b}\hat{\nabla}_y f(x_{t+1}, y_{t+1};\zeta_i) - \hat{\nabla}_y f(x_t, y_t;\zeta_i) & \text{with probability } 1-p. \end{cases}$$

then

$$\mathbb{E}\left\|v_{z,t+1} - \hat{h}_{z,t+1}\right\|^2$$

$$= \frac{p}{B}\mathbb{E}\left[\left\|\left(\sum_{i=1}^{B} D\hat{H}_v(t+1;\xi_i) - \hat{\nabla}_y f(x_{t+1}, y_{t+1};\zeta_i)\right) - \hat{h}_{z,t+1}\right\|^2\right]$$

$$+ (1-p)\left\|v_{z,t} + \sum_{i=1}^{b} D\hat{H}_v(t+1;\xi_i) - \hat{\nabla}_y f(x_{t+1}, y_{t+1};\zeta_i) - [D\hat{H}_v(t;\xi_i) - \hat{\nabla}_y f(x_t, y_t;\zeta_i)] - \hat{h}_{z,t+1} \pm \hat{h}_{z,t}\right\|^2$$

$$\leq (1-p)\mathbb{E}[\left\|v_{z,t} - \hat{h}_{z,t}\right\|^2]$$

$$+ (1-p)\mathbb{E}[\left\|\frac{1}{b}\sum_{i=1}^{b} D\hat{H}_v(t+1;\xi_i) - \hat{\nabla}_y f(x_{t+1}, y_{t+1};\zeta_i) - [D\hat{H}_v(t;\xi_i) - \hat{\nabla}_y f(x_t, y_t;\zeta_i)] - \hat{h}_{z,t+1} + \hat{h}_{z,t}\right\|^2]$$

$$\leq (1-p)\mathbb{E}[\left\|v_{z,t} - \hat{h}_{z,t}\right\|^2] + \frac{1-p}{b}\mathbb{E}[\left\|D\hat{H}_v(t+1;\xi_i) - \hat{\nabla}_y f(x_{t+1}, y_{t+1};\zeta_i) - [D\hat{H}_v(t;\xi_i) - \hat{\nabla}_y f(x_t, y_t;\zeta_i)]\right\|^2]$$

$$\leq (1-p)\mathbb{E}[\left\|v_{z,t} - \hat{h}_{z,t}\right\|^2]$$

$$+ \frac{2(1-p)}{b}\left(\mathbb{E}[\left\|D\hat{H}_v(t+1;\xi_i) - [D\hat{H}_v(t;\xi_i)]\right\|^2] + \mathbb{E}[\left\|\hat{\nabla}_y f(x_t, y_t;\zeta_i) - \hat{\nabla}_y f(x_{t+1}, y_{t+1};\zeta_i)\right\|^2]\right)$$

$$\leq (1-p)\mathbb{E}[\left\|v_{z,t} - \hat{h}_{z,t}\right\|^2] + \frac{2(1-p)}{b}\left((8r_z^2 + 3L^2)\left\|c_{t+1} - c_t\right\|^2 + 8L^2\left\|z_{t+1} - z_t\right\|^2 + 16r_z^2\rho\delta + 6\delta^2\right)$$

where the second inequality holds since we let $B = n$ in finie sum case, and $\mathbb{E}[\frac{1}{b}\sum_{i=1}^{b} D\hat{H}_v(t+1;\xi_i) - \hat{\nabla}_y f(x_{t+1}, y_{t+1};\zeta_i) - [D\hat{H}_v(t;\xi_i) - \hat{\nabla}_y f(x_t, y_t;\zeta_i)] - \hat{h}_{z,t+1} + \hat{h}_{z,t}] = 0$,

the last inequality holds due to the conclusions of lemmas D.8 and D.3, which are used to bound

$$\left[\left\|D\hat{H}_v(t+1;\xi_i) - \hat{\nabla}_y f(x_{t+1}, y_{t+1};\zeta_i) - [D\hat{H}_v(t;\xi_i) - \hat{\nabla}_y f(x_t, y_t;\zeta_i)]\right\|^2\right].$$

$$\square$$

In the following lemma, we disscuss the variance descent property of $z_t$ in Expectation case.

**Lemma E.8** (Expectation variance descent in $z$). *Under assumptions 1 and 2, we have*

$$\mathbb{E}\left\|v_{z,t+1} - \hat{h}_{z,t+1}\right\|^2 \leq (1-p)\mathbb{E}[\left\|v_{z,t} - \hat{h}_{z,t}\right\|^2]$$

$$+ \frac{2(1-p)}{b}\left((8r_z^2 + 3L^2)\left\|c_{t+1} - c_t\right\|^2 + 8L^2\left\|z_{t+1} - z_t\right\|^2 + 16r_z^2\rho\delta + 6\delta^2\right) + \frac{p}{B}(6(1+r_z^2)\sigma^2 + 48r_z^2\rho\delta + 12\delta^2).$$

*Proof.* recall that

$$v_{z,t+1} := \begin{cases} \frac{1}{B}\sum_{i=1}^{B} D\hat{H}_v(t+1;\xi_i) - \hat{\nabla}_y f(x_{t+1},y_{t+1};\zeta_i) & \text{with probability } p, \\ v_{z,t} + \frac{1}{b}\sum_{i=1}^{b}\sum D\hat{H}_v(t+1;\xi_i) - D\hat{H}_v(t;\xi_i) \\ -\frac{1}{b}\sum_{i=1}^{b}\hat{\nabla}_y f(x_{t+1},y_{t+1};\zeta_i) - \hat{\nabla}_y f(x_t,y_t;\zeta_i) & \text{with probability } 1-p. \end{cases}$$

and for the same reason as lemma E.7, but the only difference is $p\mathbb{E}[\left\|\frac{1}{B}\sum\left(D\hat{H}_v(t;\xi) - \hat{\nabla}_y f(x_t,y_t;\zeta)\right) - \hat{h}_{z,t+1}\right\|^2 \neq 0$.Therefore , we have:

$$\mathbb{E}\left\|v_{z,t+1} - \hat{h}_{z,t+1}\right\|^2$$

$$=\frac{p}{B}\mathbb{E}\left[\left\|\left(\sum_{i=1}^{B} D\hat{H}_v(t+1;\xi_i) - \hat{\nabla}_y f(x_{t+1},y_{t+1};\zeta_i)\right) - \hat{h}_{z,t+1}\right\|^2\right]$$

$$+ (1-p)\mathbb{E}[\left\|v_{z,t} - \hat{h}_{z,t}\right\|^2] + \frac{2(1-p)}{b}\left((16r_z^2+3L^2)\|c_{t+1}-c_t\|^2 + 8L^2\|z_{t+1}-z_t\|^2 + 16r_z^2\rho\delta + 6\delta^2\right)$$

$$\overset{a}{\leq}\frac{2p}{B}(3(1+r_z^2)\sigma^2 + 24r_z^2\rho\delta + 6\delta^2)$$

$$+ (1-p)\mathbb{E}[\left\|v_{z,t} - \hat{h}_{z,t}\right\|^2] + \frac{2(1-p)}{b}\left((8r_z^2+3L^2)\|c_{t+1}-c_t\|^2 + 8L^2\|z_{t+1}-z_t\|^2 + 16r_z^2\rho\delta + 6\delta^2\right),$$

where step (a) holds due to the following inequality

$$\frac{p}{B}\mathbb{E}\left[\left\|\left(\sum_{i=1}^{B} D\hat{H}_v(t+1;\xi_i) - \hat{\nabla}_y f(x_{t+1},y_{t+1};\zeta_i)\right) - \hat{h}_{z,t+1}\right\|^2\right]$$

$$\leq\frac{2p}{B}(\mathbb{E}\left[\left\|D\hat{H}_v(t;\xi_i) - D\hat{H}_v(t)\right\|^2\right] + \mathbb{E}\left[\left\|\hat{\nabla}_y f(x_t,y_t;\zeta_i) - \hat{\nabla}_y f(x_t,y_t)\right\|^2\right])$$

$$\leq\frac{2p}{B}(3(1+r_z^2)\sigma^2 + 24r_z^2\rho\delta + 6\delta^2),$$

where the last inequality holds due to the lemmas D.2 and D.7.

$\square$

### E.3 DESCENT PROPERTY IN $y$

**Lemma E.9** (Lipschitzness of $y^*$)**.** *Under assumption 1 we have*

$$\left\|y_{t+1}^* - y_t^*\right\| \leq L_{Y^*}\left\|x_{t+1} - x_t\right\|. \tag{43}$$

*where $L_{Y^*} = \kappa$*

*Proof.* from the strong convexity of $g(x,y)$ in $y$

$$L\left\|x_{t+1} - x_t\right\| \geq \left\|\nabla_y g(x_t,y_t^*) - \nabla_y g(x_{t+1},y_t^*)\right\| = \left\|\nabla_y g(x_{t+1},y_{t+1}^*) - \nabla_y g(x_{t+1},y_t^*)\right\| \geq \mu\left\|y_{t+1}^* - y_t^*\right\|,$$

where the second equality holds due the optimality of $y_t^*$ and $y_{t+1}^*$(i.e, $\nabla_y g(x_t,y_t^*)$ and $\nabla_y g(x_{t+1},y_{t+1}^*)$ =0). $\square$

**Lemma E.10.** *Under assumptions 1 and 2 we have*

$$\mathbb{E}\left[\left\|\hat{\nabla}_y g(x_t,y_t;\xi) - \hat{h}_{z,t+1}\right\|^2\right] \leq 3\sigma^2 + 6\delta^2. \tag{44}$$

*Proof.* Use the conclusion of lemma D.2 then we obtain the result directly. $\square$

In the following lemma, we disscuss the error bewteen $h_{y,t}$ and its zero-order estimator.

**Lemma E.11.** *Under assumptions 1 and 2, we have*

$$\left\| h_{y,t} - \hat{h}_{y,t} \right\|^2 \le \delta^2. \tag{45}$$

*Proof.*

$$\left\| v_{y,t} - \hat{h}_{y,t} \right\|^2 = \left\| \nabla_y g(x_t, y_t) - \hat{\nabla}_y g(x_t, y_t) \right\|^2 \le \delta^2.$$

$\square$

In the following lemma, we disscuss the inexact descent property of $y_t$.

**Lemma E.12** (Restatement of lemma 5). *Under assumption 1,let $\{y_t\}$ be a sequence generated by Algorithm 1, we have*

$$\left\| y_{t+1} - y_{t+1}^* \right\|^2 \le (1 - \frac{\eta_y \mu}{6}) \left\| y_t - y_t^* \right\|^2 - (1 - 2L\eta_y) \left\| y_{t+1} - y_t \right\|^2 + \frac{4}{\eta_y \mu} l_{Y^*}^2 \left\| x_{t+1} - x_t \right\|^2$$

$$+ \frac{4\eta_y}{\mu} (\left\| v_{y,t} - \hat{h}_{y,t} \right\|^2 + \left\| h_{y,t} - \hat{h}_{y,t} \right\|^2).$$

*Proof.* the proof is the same with lemma E.6, we omit it. $\square$

In the following lemma, we disscuss the variance descent property of variance terms in finite sum case.

**Lemma E.13** (Restatement of lemma 7). *Under assumptions 1 and 2, we have*

$$\mathbb{E} \left\| v_{y,t+1} - \hat{h}_{y,t+1} \right\|^2 \le (1 - p)\mathbb{E}[\left\| v_{y,t} - \hat{h}_{y,t} \right\|^2] + \frac{(1-p)}{b} \left( 3L^2 \left\| c_{t+1} - c_t \right\|^2 + 6\delta^2 \right).$$

*Proof.* the proof is the same with lemma E.7, we omit it. $\square$

In the following lemma, we disscuss the variance descent property of $y_t$ in Expectation case.

**Lemma E.14** (variance of $y$ in Expectation case). *Under assumptions 1 and 2, we have*

$$\mathbb{E} \left\| v_{y,t+1} - \hat{h}_{y,t+1} \right\|^2 \le (1 - p)\mathbb{E}[\left\| v_{y,t} - \hat{h}_{y,t} \right\|^2] + \frac{(1-p)}{b} \left( 3L^2 \left\| c_{t+1} - c_t \right\|^2 + 6\delta^2 \right) + \frac{p}{B} \left( 6\delta^2 + 3\sigma^2 \right)$$

*Proof.* the proof is the same with lemma E.8, we omit it. $\square$

### E.4 DESCENT IN $x$

**Lemma E.15** (Restatement of lemma 1). *Under assumption 1, for the error between hypergradient $\Phi(x_t)$ and approximation $h_{x,t}$ in (2), we have the following upper bound,*

$$\left\| h_{x,t} - \nabla \Phi(x_t) \right\|^2 \le (2L^2 + 4r_z^2 \rho^2) \left\| y_t - y^* \right\|^2 + 4L^2 \left\| z_t - z_t^* \right\|^2 \tag{46}$$

*Proof.* $\left\| h_{x,t} - \nabla \Phi(x_t) \right\|^2$:

$$\left\| h_{x,t} - \nabla \Phi(x_t) \right\|^2$$
$$\le 2 \left\| \nabla_x f(x_t, y_t) - \nabla_x f(x_t, y_t^*) \right\|^2 + 2 \left\| \nabla_{xy}^2 g(x_t, y_t) z_t - \nabla_{xy}^2 g(x_t, y_t^*) z_t^* \right\|^2$$
$$\le 2L^2 \left\| y_t - y_t^* \right\|^2 + 4(\left\| z_t \right\|^2 \left\| \nabla_{xy}^2 g(x_t, y_t) - \nabla_{xy}^2 g(x_t, y_t^*) \right\|^2 + \left\| \nabla_{xy}^2 g(x_t, y_t^*) \right\|^2 \left\| z_t - z_t^* \right\|^2)$$
$$\le (2L^2 + 4r_z^2 \rho^2) \left\| y_t - y^* \right\|^2 + 4L^2 \left\| z_t - z_t^* \right\|^2.$$

$\square$

**Lemma E.16.** *Under assumptions 1 and 2, for the variance of sample zeroth-order estimator, we have*

$$\mathbb{E}\left[\left\|\left(D\hat{J}_v(t;\xi) - \hat{\nabla}_x f(x_t, y_t; \zeta)\right) - \hat{h}_{x,t+1}\right\|^2\right] \le 6(1 + r_z^2)\sigma^2 + 12r_z^2\rho\delta + 6\delta^2 \quad (47)$$

*Proof.*

$$\mathbb{E}\left[\left\|\left(D\hat{J}_v(t;\xi) - \hat{\nabla}_y f(x_t, y_t; \zeta)\right) - \hat{h}_{x,t+1}\right\|^2\right]$$

$$\le 2\mathbb{E}\left[\left\|D\hat{J}_v(t;\xi) - D\hat{J}_v(t)\right\|^2\right] + 2\mathbb{E}\left[\left\|\hat{\nabla}_x f(x_t, y_t; \zeta) - \hat{\nabla}_x f(x_t, y_t)\right\|^2\right]$$

$$\le 2(3r_z^2\sigma^2 + 24r_z^2\rho\delta + 6\delta^2 + 3\sigma^2)$$

$$= 6(1 + r_z^2)\sigma^2 + 48r_z^2\rho\delta + 12\delta^2$$

where the last inequality holds due to the conclusions of D.7 and D.2, which are used to upper bound $\mathbb{E}\left[\left\|D\hat{J}_v(t;\xi) - D\hat{J}_v(t)\right\|^2\right]$ and $\mathbb{E}\left[\left\|\hat{\nabla}_x f(x_t, y_t; \zeta) - \hat{\nabla}_x f(x_t, y_t)\right\|^2\right]$. $\qquad\square$

**Lemma E.17** (Smoothness of function $\Phi(x)$ )**.** *we have*

$$\|\Phi(x_1) - \Phi(x_2)\| \le L_\phi \|x_1 - x_2\|. \quad (48)$$

*where $L_\phi = \mathcal{O}(\kappa^3)$.*

*See Lemma 2.2 in (Ghadimi and Wang, 2018) for detail proof.*

**Lemma E.18** (Restatement of lemma 8)**.** *Under assumptions 1 and 2, for Algorithm 1, we have:*

$$\Phi(x_{t+1}) \le \Phi(x_t) - \frac{\eta_x}{2}\|\nabla\Phi(x_t)\|^2 - \left(\frac{1}{2\eta_x} - \frac{L_\phi}{2}\right)\|x_{t+1} - x_t\|^2 + \frac{3\eta_x}{2}\|v_{x,t} - \hat{h}_{x,t}\|^2$$

$$+ \frac{3\eta_x}{2}((2L^2 + 4r_z^2\rho^2)\|y_t - y^*\|^2 + 4L^2\|z_t - z_t^*\|^2 + 2\delta^2 + 8r_z^2\rho\delta),$$

*Proof.* from the conclusion of lemma C.7, we have

$$\Phi(x_{t+1}) \le \Phi(x_t) - \frac{\eta_x}{2}\|\nabla\Phi(x_t)\|^2 - \left(\frac{1}{2\eta_x} - \frac{L_\phi}{2}\right)\|x_{t+1} - x_t\|^2 + \frac{\eta_x}{2}\|v_{x,t} - \nabla\Phi(x_t)\|^2$$

$$\le \Phi(x_t) - \frac{\eta_x}{2}\|\nabla\Phi(x_t)\|^2 - \left(\frac{1}{2\eta_x} - \frac{L_\phi}{2}\right)\|x_{t+1} - x_t\|^2$$

$$+ \frac{3\eta_x}{2}(\|v_{x,t} - \hat{h}_{x,t}\|^2 + \|\hat{h}_{x,t} - h_{x,t}\|^2 + \|h_{x,t} - \nabla\Phi(x_t)\|^2)$$

$$\le \Phi(x_t) - \frac{\eta_x}{2}\|\nabla\Phi(x_t)\|^2 - \left(\frac{1}{2\eta_x} - \frac{L_\phi}{2}\right)\|x_{t+1} - x_t\|^2 + \frac{3\eta_x}{2}\|v_{x,t} - \hat{h}_{x,t}\|^2$$

$$+ \frac{3\eta_x}{2}((2L^2 + 4r_z^2\rho^2)\|y_t - y^*\|^2 + 4L^2\|z_t - z_t^*\|^2 + 2\delta^2 + 8r_z^2\rho\delta),$$

here, to obtain the last inequality, we bound $\|h_{x,t} - \nabla\Phi(x_t)\|^2$ by using lemma E.15 and $\|\hat{h}_{x,t} - h_{x,t}\|^2$ is bounded by the following fact:

$$\|\hat{h}_{x,t} - h_{x,t}\|^2 \le 2\delta^2 + 2\left\|D\hat{H}_v(t) - H_v(t)\right\|^2$$

$$\overset{\text{lemma } D.6}{\le} 2\delta^2 + 8r_z^2\rho\delta.$$

$\qquad\square$

In the following lemma, we discuss the inexact descent property of variance term in finite sum case.

**Lemma E.19** (Restatement of lemma 9). *Under assumptions 1 and 2, we have*

$$\mathbb{E}\left\|v_{x,t+1} - \hat{h}_{x,t+1}\right\|^2$$

$$\leq (1-p)\mathbb{E}[\left\|v_{x,t} - \hat{h}_{x,t}\right\|^2] + \frac{2(1-p)}{b}\left((8r_z^2 + 3L^2)\left\|c_{t+1} - c_t\right\|^2 + 8L^2\left\|z_{t+1} - z_t\right\|^2 + 16r_z^2\rho\delta + 6\delta^2\right).$$

*Proof.* recall that

$$v_{x,t+1} := \begin{cases} \frac{1}{B}\sum_{i=1}^{B}\hat{\nabla}_x f(x_{t+1}, y_{t+1}; \zeta_i) - D\hat{J}_v(t+1; \xi_i) & \text{with probability } p, \\ v_{x,t+1} + \frac{1}{b}\sum_{i=1}^{b}\hat{\nabla}_x f(x_{t+1}, y_{t+1}; \zeta_i) - \hat{\nabla}_x f(x_t, y_t; \zeta_i) \\ -\frac{1}{b}\sum_{i=1}^{b} D\hat{J}_v(t+1; \xi_i) - D\hat{J}_v(t; \xi_i) & \text{with probability } 1-p. \end{cases}$$

then

$$\mathbb{E}\left\|v_{x,t+1} - \hat{h}_{x,t+1}\right\|^2$$

$$= p\mathbb{E}\left\|\left(\frac{1}{B}\sum_{i=1}^{B}\hat{\nabla}_x f(x_{t+1}, y_{t+1}; \zeta_i) - D\hat{J}_v(t+1; \xi_i)\right) - \hat{h}_{x,t+1}\right\|^2$$

$$+ (1-p)\mathbb{E}\left\|v_{x,t} + \frac{1}{b}\sum_{i=1}^{b}\hat{\nabla}_x f(x_{t+1}, y_{t+1}; \zeta_i) - D\hat{J}_v(t; \xi_i) - [\hat{\nabla}_x f(x_t, y_t; \zeta) - D\hat{J}_v(t; \xi_i)] - \hat{h}_{x,t+1} \pm \hat{h}_{x,t}\right\|^2]$$

$$\leq (1-p)\mathbb{E}[\left\|v_{x,t} - \hat{h}_{x,t}\right\|^2]$$

$$+ (1-p)\mathbb{E}[\left\|\frac{1}{b}\sum_{i=1}^{b}\hat{\nabla}_x f(x_{t+1}, y_{t+1}; \zeta_i) - D\hat{J}_v(t+1; \xi_i) - [\hat{\nabla}_x f(x_t, y_t; \zeta_i) - D\hat{J}_v(t; \xi_i)] - \hat{h}_{x,t+1} + \hat{h}_{x,t}\right\|^2]$$

$$\leq (1-p)\mathbb{E}[\left\|v_{x,t} - \hat{h}_{x,t}\right\|^2] + \frac{1-p}{b}\mathbb{E}[\left\|f(x_{t+1}, y_{t+1}; \zeta_i) - D\hat{J}_v(t+1; \xi_i) - [\hat{\nabla}_x f(x_t, y_t; \zeta_i) - D\hat{J}_v(t; \xi_i)]\right\|^2]$$

$$\leq (1-p)\mathbb{E}[\left\|v_{x,t} - \hat{h}_{x,t}\right\|^2] + \frac{2(1-p)}{b}\left((8r_z^2 + 3L^2)\left\|c_{t+1} - c_t\right\|^2 + 8L^2\left\|z_{t+1} - z_t\right\|^2 + 16r_z^2\rho\delta + 6\delta^2\right),$$

where the second inequality holds since we let $B = n$ in finie sum case, and $\mathbb{E}[\frac{1}{b}\sum_{i=1}^{b}\hat{\nabla}_x f(x_{t+1}, y_{t+1}; \zeta_i) - D\hat{J}_v(t+1; \xi_i) - [\hat{\nabla}_x f(x_t, y_t; \zeta_i) - D\hat{J}_v(t; \xi_i)] - \hat{h}_{x,t+1} + \hat{h}_{x,t}] = 0$, the last inequality holds due to the conclusions of lemmas D.8 and D.3, which are used to bound $\mathbb{E}[\left\|f(x_{t+1}, y_{t+1}; \zeta_i) - D\hat{J}_v(t+1; \xi_i) - [\hat{\nabla}_x f(x_t, y_t; \zeta_i) - D\hat{J}_v(t; \xi_i)]\right\|^2]$. $\square$

In the following lemma, we disscuss the variance descent property in Expectation case.

**Lemma E.20** (Expectation variance descent in $x$). *Under assumptions 1 and 2, we have*

$$\mathbb{E}\left\|v_{x,t+1} - \hat{h}_{x,t+1}\right\|^2 \leq (1-p)\mathbb{E}[\left\|v_{x,t} - \hat{h}_{x,t}\right\|^2]$$

$$+ \frac{2(1-p)}{b}\left((8r_z^2 + 3L^2)\left\|c_{t+1} - c_t\right\|^2 + 8L^2\left\|z_{t+1} - z_t\right\|^2 + 16r_z^2\rho\delta + 6\delta^2\right) + \frac{p}{B}(6(1+r_z^2)\sigma^2 + 48r_z^2\rho\delta + 12\delta^2).$$

*Proof.* recall that

$$v_{x,t+1} := \begin{cases} \frac{1}{B}\sum_{i=1}^{B}\hat{\nabla}_x f(x_{t+1}, y_{t+1}; \zeta_i) - D\hat{J}_v(t+1; \xi_i) & \text{with probability } p, \\ v_{x,t+1} + \frac{1}{b}\sum_{i=1}^{b}\hat{\nabla}_x f(x_{t+1}, y_{t+1}; \zeta_i) - \hat{\nabla}_x f(x_t, y_t; \zeta_i) \\ -\frac{1}{b}\sum_{i=1}^{b} D\hat{J}_v(t+1; \xi_i) - D\hat{J}_v(t; \xi_i) & \text{with probability } 1-p. \end{cases}$$

then

$$\mathbb{E} \left\| v_{x,t+1} - \hat{h}_{x,t+1} \right\|^2$$

$$= p\mathbb{E} \left\| \left( \frac{1}{B} \sum_{i=1}^{B} \hat{\nabla}_x f(x_{t+1}, y_{t+1}; \zeta_i) - D\hat{J}_v(t+1; \xi_i) \right) - \hat{h}_{x,t+1} \right\|^2$$

$$+ (1-p)\mathbb{E}[\left\| v_{x,t} - \hat{h}_{x,t} \right\|^2] + \frac{2(1-p)}{b} \left( (8r_z^2 + 3L^2) \left\| c_{t+1} - c_t \right\|^2 + 8L^2 \left\| z_{t+1} - z_t \right\|^2 + 16r_z^2\rho\delta + 6\delta^2 \right)$$

$$\overset{a}{\leq} \frac{2p}{B}(3(1+r_z^2)\sigma^2 + 24r_z^2\rho\delta + 6\delta^2)$$

$$+ (1-p)\mathbb{E}[\left\| v_{x,t} - \hat{h}_{x,t} \right\|^2] + \frac{2(1-p)}{b} \left( (8r_z^2 + 3L^2) \left\| c_{t+1} - c_t \right\|^2 + 8L^2 \left\| z_{t+1} - z_t \right\|^2 + 16r_z^2\rho\delta + 6\delta^2 \right),$$

where step (a) hold due to the following inequality:

$$p\mathbb{E}[\left\| \frac{1}{B} \sum_{i=1}^{B} \hat{\nabla}_x f(x_t, y_t; \zeta_i) - D\hat{J}_v(t; \xi_i) - \hat{h}_{z,t+1} \right\|^2$$

$$\leq \frac{2p}{B}(\mathbb{E}\left[\left\| D\hat{J}_v(t; \xi_i) - D\hat{J}_v(t) \right\|^2 \right] + \mathbb{E}\left[ \left\| \hat{\nabla}_x f(x_t, y_t; \zeta) - \hat{\nabla}_x f(x_t, y_t) \right\|^2 \right])$$

$$\leq \frac{2p}{B}(3(1+r_z^2)\sigma^2 + 24r_z^2\rho\delta + 6\delta^2),$$

where the last inequality due to the conclusion of lemmas D.2 and D.7.

$$\square$$

### E.5 PROOF SKETCH OF THEOREM 1

The main goal of our proof is to bound potential function, starting from the inexact gradient descent of $\Phi(x)$ in Lemma 8, which is a key step to estimate the potential function.

$$\Phi(x_{t+1}) \leq \Phi(x_t) - \frac{\eta_x}{2} \|\nabla\Phi(x_t)\|^2 - \left( \frac{1}{2\eta_x} - \frac{L_\phi}{2} \right) \|x_{t+1} - x_t\|^2 + \mathbb{E}[\frac{3\eta_x}{2} \|v_{x,t} - \hat{h}_{x,t}\|^2]$$

$$+ c_1\eta_x \|y_t - y^*\|^2 + c_2\eta_x \|z_t - z_t^*\|^2) + \text{other terms} \quad \text{for some constant } c_1, c_2.$$

Observing the following descent of $z_t$ and $y_t$ and variance term $\|v_{x,t} - \hat{h}_{x,t}\|^2$ in Lemmas 4, 5 and 9:

$$\left\| y_{t+1} - y_{t+1}^* \right\|^2 - \left\| y_t - y_t^* \right\|^2 \leq -\frac{\eta_y\mu}{6} \left\| y_t - y_t^* \right\|^2 + \frac{4\eta_y}{\mu}(\left\| v_{y,t} - \hat{h}_{y,t} \right\|^2) + \text{other terms} ,$$

$$\left\| z_{t+1} - z_{t+1}^* \right\|^2 - \left\| z_t - z_t^* \right\|^2 \leq -\frac{4\eta_z\mu}{6} \left\| z_t - z_t^* \right\|^2 + \frac{4\eta_z}{\mu}(\left\| v_{z,t} - \hat{h}_{z,t} \right\|^2) + \text{other terms} ,$$

and $\mathbb{E} \left\| v_{x,t+1} - \hat{h}_{x,t+1} \right\|^2 - \mathbb{E}[\left\| v_{x,t} - \hat{h}_{x,t} \right\|^2] \leq -p\mathbb{E}[\left\| v_{x,t} - \hat{h}_{x,t} \right\|^2] + \text{other terms} .$

Next, we define a potential function $\Psi_t$ as follows:

$$\Psi_t := \Phi(x_t) + \mathbb{E}[\frac{3\eta_x}{2p} \|v_{x,t} - \hat{h}_{x,t}\|^2] + \frac{c_1}{\frac{\eta_y\mu}{6}} \left( \|y_t - y_t^*\|^2 \right) + \frac{c_2}{\frac{\eta_z\mu}{6}} \left( \|z_t - z_t^*\|^2 \right)$$

then by simple calculation, we can obtain the following results for some constants $c_3, c_4$:

$$\Psi_{t+1} - \Psi_t \leq \Phi(x_{t+1}) - \Phi(x_t) - \frac{3\eta_x}{2}\mathbb{E}[\|v_{x,t} - \hat{h}_{x,t}\|^2] - c_1\eta_x \|y_t - y^*\|^2 - c_2\eta_x \|z_t - z_t^*\|^2$$

$$+ c_3(\|v_{y,t} - \hat{h}_{y,t}\|^2) + c_4(\|v_{z,t} - \hat{h}_{z,t}\|^2) + \text{other terms}$$

we can see that the negative terms of $\Psi_{t+1} - \Psi_t$ can precisely cancels out the positive terms of $\Phi(x_{t+1}) - \Phi(x_t)$, we further get:

$$\Psi_{t+1} - \Psi_t \leq c_3\|v_{y,t} - \hat{h}_{y,t}\|^2 + c_4\|v_{z,t} - \hat{h}_{z,t}\|^2 + \text{other terms} ,$$

similarly , from the descent of variance term in Lemmas 6 and 7 as follows:

$$\mathbb{E}\|v_{z,t+1} - \hat{h}_{z,t+1}\|^2 - \mathbb{E}[\|v_{z,t} - \hat{h}_{z,t}\|^2] \leq -p\mathbb{E}[\|v_{z,t} - \hat{h}_{z,t}\|^2] + \text{other terms} .$$

$$\mathbb{E}\|v_{y,t+1} - \hat{h}_{y,t+1}\|^2 - \mathbb{E}[\|v_{y,t} - \hat{h}_{y,t}\|^2] \leq -p\mathbb{E}[\|v_{y,t} - \hat{h}_{y,t}\|^2] + \text{other terms} .$$

we define the second potential function $\psi_t$ as follows:

$$\psi_t := \Psi_t + \frac{c_3}{p}\mathbb{E}[\|v_{y,t} - \hat{h}_{y,t}\|^2] + \frac{c_4}{p}\mathbb{E}[\|v_{z,t} - \hat{h}_{z,t}\|^2],$$

Similarly, the negative terms of $\psi_{t+1} - \psi_t$ can precisely cancel out the positive variance terms of $\Psi_{t+1} - \Psi_t$. After that, by calculating the "other terms", we get an equation like this for some constants $c_5, c_6, c_7, c_8$:

$$\psi_{t+1} \leq \psi_t - \frac{\eta_x}{2}\|\nabla\Phi(x_t)\|^2 + c_5\mathcal{O}(\delta) + \eta_x\left(c_6\|x_{t+1} - x_t\|^2 + c_7\|y_{t+1} - y_t\|^2 + c_8\|z_{t+1} - z_t\|^2\right),$$

by the choice of $\eta_x, \eta_y, \eta_z$, we let $c_6, c_7, c_8 \leq 0, c_5\delta \leq \eta_x\mathcal{O}(\epsilon^2)$. We will obtain the final result in Theorem 1:

$$\psi_{t+1} \leq \psi_t - \frac{\eta_x}{2}\left(\|\nabla\Phi(x_t)\|^2 - \epsilon^2\right).$$

**Theorem E.1** (Restatement of Theorem 1). *Define Lyapunov function*

$$\psi_t := \Phi(x_t) + \frac{18(L^2 + 2r_z^2\rho^2)\eta_x}{\eta_y\mu}\left(\|y_t - y_t^*\|^2\right) + \frac{36\eta_x L^2\rho^2}{\eta_z\mu}\left(\|z_t - z_t^*\|^2\right)$$

$$+ \frac{3\eta_x}{2p}\|v_{x,t} - \hat{h}_{x,t}\|^2 + \frac{72(L^2 + 2r_z^2\rho^2)\eta_x}{p\mu^2}\left\|v_{y,t} - \hat{h}_{y,t}\right\|^2 + \frac{144L^2\rho^2\eta_x}{p\mu^2}\left\|v_{z,t} - \hat{h}_{z,t}\right\|^2,$$

*under assumptions 1 and 2, for Algorithm 1, choose* $\eta_y \leq \frac{1}{2\kappa}, \eta_z \leq \frac{1}{L_{\max}\kappa}, \eta_x \leq \frac{1}{\mathcal{O}(L_{\max}^2\kappa^4)}, p = \frac{1}{\sqrt{n}},$
$b = \sqrt{n}, B = n, v \leq \frac{\epsilon^2}{\sqrt{d_1 + d_2}\kappa^4 L_{\max}^5}$ *to let* $\delta \leq \frac{\epsilon^2}{\mathcal{O}(L_{\max}^5\kappa^4)}, h = \frac{2}{r_z}\sqrt{\frac{\delta}{\rho}},$ *we obtain:*

$$\mathbb{E}[\psi_{t+1} - \psi_t] \leq -\mathbb{E}[\frac{\eta_x}{2}(\|\nabla\Phi(x_t)\|^2 - 2\epsilon^2)],$$

*and total oracle cost is*

$$\#funtion = dT(pn + b) + dn = \mathcal{O}((d_1 + d_2)\sqrt{n}\epsilon^{-2}L_{\max}^2\kappa^4 + (d_1 + d_2)n).$$

*Proof.*

$$\Phi(x_{t+1}) \leq \Phi(x_t) - \frac{\eta_x}{2}\|\nabla\Phi(x_t)\|^2 - \left(\frac{1}{2\eta_x} - \frac{L_\phi}{2}\right)\|x_{t+1} - x_t\|^2 + \frac{3\eta_x}{2}\|v_{x,t} - \hat{h}_{x,t}\|^2$$

$$+ \frac{3\eta_x}{2}((2L^2 + 4r_z^2\rho^2)\|y_t - y^*\|^2 + 4L^2\|z_t - z_t^*\|^2 + 2\delta^2 + 8r_z^2\rho\delta),$$

define Lyapunov function

$$\Psi_t := \Phi(x_t) + \frac{3\eta_x}{2p}\mathbb{E}[\|v_{x,t} - \hat{h}_{x,t}\|^2] + \frac{72(L^2 + 2r_z^2\rho^2)\eta_x}{\eta_y\mu}\left(\|y_t - y_t^*\|^2\right) + \frac{144\eta_x L^2\rho^2}{\eta_z\mu}\left(\|z_t - z_t^*\|^2\right),$$

by the definition of $\Psi_t$ and lemmas E.6, E.12, E.18, E.19, we have

$$\mathbb{E}[\Psi_{t+1} - \Psi_t]$$

$$\leq \mathbb{E}[-\frac{\eta_x}{2}\|\nabla\Phi(x_t)\|^2 - \left(\frac{1}{2\eta_x} - \frac{L_\phi}{2}\right)\|x_{t+1} - x_t\|^2 + \frac{3\eta_x}{2}(2\delta^2 + 8r_z^2\rho\delta)]$$

$$+ \mathbb{E}[\frac{36L^2\rho^2\eta_x}{\eta_z\mu}\left(-(1 - L\eta_z)\|z_{t+1} - z_t\|^2 + \frac{4}{\eta_z\mu}l_{Z^*}^2\|c_{t+1} - c_t\|^2 + \frac{4\eta_z}{\mu}(\left\|v_{z,t} - \hat{h}_{z,t}\right\|^2 + \left\|h_{z,t} - \hat{h}_{z,t}\right\|^2)\right)]$$

$$+ \mathbb{E}[\frac{18(L^2 + 2r_z^2\rho^2)\eta_x}{\eta_y\mu}\left(-(1 - 2L\eta_y)\|y_{t+1} - y_t\|^2 + \frac{4}{\eta_y\mu}l_{Y^*}^2\|x_{t+1} - x_t\|^2 + \frac{4\eta_y}{\mu}(\left\|v_{y,t} - \hat{h}_{y,t}\right\|^2 + \left\|h_{y,t} - \hat{h}_{y,t}\right\|^2)\right)]$$

$$+ \mathbb{E}[\frac{3(1 - p)\eta_x}{bp}\left((8r_z^2 + 3L^2)\|c_{t+1} - c_t\|^2 + 8L^2\|z_{t+1} - z_t\|^2 + 16r_z^2\rho\delta + 6\delta^2\right)],$$

Then, let $\frac{1-p}{bp} = 1$, use the conclusions of lemmas E.11 and E.3 to upper bound $\left\| h_{y,t} - \hat{h}_{y,t} \right\|^2$, $\left\| h_{z,t} - \hat{h}_{z,t} \right\|^2$, and rearrange the terms, we obtain:

$$\mathbb{E}[\Psi_{t+1} - \Psi_t]$$
$$\leq \mathbb{E}[-\frac{\eta_x}{2} \|\nabla\Phi(x_t)\|^2 - \left(\frac{1}{2\eta_x} - \frac{L_\phi}{2}\right) \|x_{t+1} - x_t\|^2]$$
$$+ \eta_x \left( \left(21 + \frac{72(L^2 + 2r_z^2\rho^2)}{\eta_y\mu} + \frac{288L^2\rho^2}{\eta_z\mu}\right)\delta^2 + \left(60 + \frac{864L^2\rho^2}{\eta_z\mu}\right)r_z^2\rho\delta \right)$$
$$+ \mathbb{E}[\frac{36L^2\rho^2\eta_x}{\eta_z\mu}\left(-(1 - L\eta_z)\|z_{t+1} - z_t\|^2 + \frac{4}{\eta_z\mu}l_{Z^*}^2\|c_{t+1} - c_t\|^2\right) + \frac{144L^2\rho^2\eta_x}{\mu^2}\left\| v_{z,t} - \hat{h}_{z,t} \right\|^2]$$
$$+ \mathbb{E}[\frac{18(L^2 + 2r_z^2\rho^2)\eta_x}{\eta_y\mu}\left(-(1 - 2L\eta_y)\|y_{t+1} - y_t\|^2 + \frac{4}{\eta_y\mu}l_{Y^*}^2\|x_{t+1} - x_t\|^2\right) + \frac{72(L^2 + 2r_z^2\rho^2)\eta_x}{\mu^2}\left\| v_{y,t} - \hat{h}_{y,t} \right\|^2]$$
$$+ \mathbb{E}[\frac{3(1-p)\eta_x}{bp}\left((8r_z^2 + 3L^2)\|c_{t+1} - c_t\|^2 + 8L^2\|z_{t+1} - z_t\|^2\right)],$$

denote Lyapunov function $\psi_t = \Psi_t + \frac{72(L^2 + 2r_z^2\rho^2)\eta_x}{p\mu^2}\left\| v_{y,t} - \hat{h}_{y,t} \right\|^2 + \frac{144L^2\rho^2\eta_x}{p\mu^2}\left\| v_{z,t} - \hat{h}_{z,t} \right\|^2$, and use lemma E.7 and E.13 to bound $\left\| v_{y,t} - \hat{h}_{y,t} \right\|^2$, $\left\| v_{z,t} - \hat{h}_{z,t} \right\|^2$, we have

$$\mathbb{E}[\psi_{t+1} - \psi_t]$$
$$\leq \mathbb{E}[-\frac{\eta_x}{2} \|\nabla\Phi(x_t)\|^2 - \left(\frac{1}{2\eta_x} - \frac{L_\phi}{2}\right)\|x_{t+1} - x_t\|^2]$$
$$+ \eta_x \left(21 + \frac{72(L^2 + 2r_z^2\rho^2)}{\eta_y\mu} + \frac{288L^2\rho^2}{\eta_z\mu} + \frac{432(L^2 + 2r_z^2\rho^2)}{\mu^2} + \frac{864L^2\rho^2}{\mu^2}\right)\delta^2$$
$$+ \eta_x \left(60 + \frac{864L^2\rho^2}{\eta_z\mu} + \frac{2304L^2\rho^2}{\mu^2}\right)r_z^2\rho\delta$$
$$+ \mathbb{E}[\frac{144L^2\rho^2\eta_x}{\eta_z\mu}\left(-(1 - L\eta_z)\|z_{t+1} - z_t\|^2 + \frac{4}{\eta_z\mu}l_{Z^*}^2\|c_{t+1} - c_t\|^2\right)]$$
$$+ \mathbb{E}[\frac{72(L^2 + 2r_z^2\rho^2)\eta_x}{\eta_y\mu}\left(-(1 - 2L\eta_y)\|y_{t+1} - y_t\|^2 + \frac{4}{\eta_y\mu}l_{Y^*}^2\|x_{t+1} - x_t\|^2\right)]$$
$$+ \mathbb{E}[3\eta_x\left((8r_z^2 + 3L^2)\|c_{t+1} - c_t\|^2 + 8L^2\|z_{t+1} - z_t\|^2\right)]$$
$$+ \mathbb{E}[\frac{72(L^2 + 2r_z^2\rho^2)\eta_x}{\mu^2}\left(3L^2\|c_{t+1} - c_t\|^2\right) + \frac{288L^2\rho^2\eta_x}{\mu^2}\left((8r_z^2 + 3L^2)\|c_{t+1} - c_t\|^2 + 8L^2\|z_{t+1} - z_t\|^2\right)],$$

rearrange terms we obtain:

$$\mathbb{E}[\psi_{t+1} - \psi_t]$$

$$\leq -\frac{\eta_x}{2}\|\nabla\Phi(x_t)\|^2$$

$$+ \underbrace{\eta_x\left(21 + \frac{72(L^2 + 2r_z^2\rho^2)}{\eta_y\mu} + \frac{288L^2\rho^2}{\eta_z\mu} + \frac{432(L^2 + 2r_z^2\rho^2)}{\mu^2} + \frac{864L^2\rho^2}{\mu^2}\right)\delta^2}_{C_{\delta_1}}$$

$$+ \underbrace{\eta_x\left(60 + \frac{864L^2\rho^2}{\eta_z\mu} + \frac{2304L^2\rho^2}{\mu^2}\right)r_z^2\rho\delta}_{C_{\delta_2}}$$

$$- \mathbb{E}\left[\left(\frac{1}{2\eta_x} - \frac{L_\phi}{2} - \eta_x\left(\frac{288(L^2 + 2r_z^2\rho^2)l_{Y^*}^2}{\eta_y^2\mu^2} + \frac{576L^2\rho^2l_{Z^*}^2}{\eta_z^2\mu^2}\right)\right.\right.$$

$$\left.\left.- \eta_x\left(24r_z^2 + 9L^2 + \frac{216(L^4 + 2L^2r_z^2\rho^2)}{\mu^2} + \frac{288L^2\rho^2(8r_z^2 + 3L^2)}{\mu^2}\right)\right)\|x_{t+1} - x_t\|^2\right]$$

$$- \mathbb{E}\left[\eta_x\left(\frac{144L^2\rho^2}{\eta_z\mu} - \frac{144L^3\rho^2}{\mu} - 24L^2 - \frac{2304L^4\rho^2}{\mu^2}\right)\|z_{t+1} - z_t\|^2\right]$$

$$- \mathbb{E}\left[\left(\frac{72(L^2 + 2r_z^2\rho^2)}{\eta_y\mu} - \frac{72(L^3 + 2Lr_z^2\rho^2)}{\mu}\right.\right.$$

$$\left.\left.- \eta_x\left(\frac{576L^2\rho^2}{\eta_z^2\mu^2} + (24r_z^2 + 9L^2) + \frac{216(L^4 + 2L^2r_z^2\rho^2)}{\mu^2} + \frac{288L^2\rho^2(8r_z^2 + 3L^2)}{\mu^2}\right)\right)\|y_{t+1} - y_t\|^2\right],$$

let's discuss the how to choose step size to let the coefficient of $\|x_{t+1} - x_t\|^2, \|y_{t+1} - y_t\|^2, \|z_{t+1} - z_t\|^2 \leq 0$, to simplify denote the coefficient of $\|x_{t+1} - x_t\|^2, \|y_{t+1} - y_t\|^2, \|z_{t+1} - z_t\|^2$ be $C_x, C_y, C_z$, we have

- $\|z_{t+1} - z_t\|^2$: let $\eta_z \leq \frac{144L^2\rho^2}{\mu(\frac{144L^3\rho^2}{\mu} + 24L^2 + \frac{2304L^4\rho^2}{\mu^2})} = \mathcal{O}(\frac{1}{L_{\max}\kappa})$, we obtain $C_z \leq 0$

- $\|y_{t+1} - y_t\|^2$: let $\eta_y \leq \frac{1}{2\kappa}$, we obtain

$$C_y = -\mathcal{O}\left(\frac{L_{max}\kappa^3}{\eta_y} - \eta_x\mathcal{O}(L^4\kappa^4)\right)$$

  we further let $\eta_y \leq \frac{1}{\eta_x\mathcal{O}(L_{\max}^3\kappa)}$, thus we can obtain $C_y \leq 0$.

- $\|x_{t+1} - x_t\|^2$: let $\eta_x = \frac{1}{2L_\phi}$, denote coefficient of $\|x_{t+1} - x_t\|^2$ as $C_x$, we have

$$C_x \leq -\left(\frac{1}{4\eta_x} - \eta_x\mathcal{O}\left(\frac{\kappa^6}{\eta_y^2} + L^4\kappa^8\right)\right)$$

  choose $\eta_x \leq \min\{\frac{\eta_y}{L_{\max}^{1.5}}, \frac{1}{L_{\max}^2\kappa^4}\}$, we have $C_x \leq 0$, and we can easily validate that choosing $\eta_y = \frac{1}{2\kappa}, \eta_x = \mathcal{O}(\frac{1}{L^2\kappa^4})$ can simultaneously satisfy both $\eta_x \leq \min\{\frac{\eta_y}{L^{1.5}}, \frac{1}{L^2\kappa^4}\}$ and $\eta_y \leq \frac{1}{\eta_x\mathcal{O}(L_{\max}^3\kappa)}$.

since $C_x, C_y, C_z \leq 0$, choose $\delta$ suffciently small, i.e, $\delta \leq \frac{\epsilon^2}{\mathcal{O}(L_{\max}^5\kappa^4)}$ to let $C_{\delta_1} \leq \eta_x\epsilon^2, C_{\delta_2} \leq \eta_x\epsilon^2$, we obtain:

$$\mathbb{E}[\psi_{t+1} - \psi_t] \leq -\mathbb{E}[\frac{\eta_x}{2}(\|\nabla\Phi(x_t)\|^2 - 2\epsilon^2)],$$

thus $T = \frac{\epsilon^{-2}}{\eta_x} = \mathcal{O}(\epsilon^{-2}L_{\max}^2\kappa^4)$ and total oracle cost is

$$\#funtion = dT(pn + b) + dn = \mathcal{O}((d_1 + d_2)\sqrt{n}\epsilon^{-2}L_{\max}^2\kappa^4 + (d_1 + d_2)n).$$

$\square$

**Theorem E.2** (Restatement of Theorem 2). *Define Lyapunov function*

$$\psi_t := \Phi(x_t) + \frac{18(L^2 + 2r_z^2\rho^2)\eta_x}{\eta_y\mu}\left(\|y_t - y_t^*\|^2\right) + \frac{36\eta_x L^2\rho^2}{\eta_z\mu}\left(\|z_t - z_t^*\|^2\right)$$

$$+ \frac{3\eta_x}{2p}\|v_{x,t} - \hat{h}_{x,t}\|^2 + \frac{72(L^2 + 2r_z^2\rho^2)\eta_x}{p\mu^2}\left\|v_{y,t} - \hat{h}_{y,t}\right\|^2 + \frac{144L^2\rho^2\eta_x}{p\mu^2}\left\|v_{z,t} - \hat{h}_{z,t}\right\|^2,$$

*choose* $\eta_y \le \frac{1}{2\kappa}, \eta_z \le \frac{1}{L\kappa}, \eta_x \le \frac{1}{\mathcal{O}(L^2\kappa^4)}$, $p = \frac{\epsilon}{\sigma\kappa^2}$, $b = \sigma\epsilon^{-1}\kappa^2$, $v \le \frac{\epsilon^2}{\sqrt{d_1 + d_2}\kappa^4 L_{\max}^5}$ *to let*

$\delta \le \frac{\epsilon^2}{\mathcal{O}(L_{\max}^5\kappa^4)}$, $B \ge \mathcal{O}(L_{\max}^2\kappa^4\epsilon^{-2}\sigma^2)$, *choose* $h = \frac{2}{r_z}\sqrt{\frac{\delta}{\rho}}$, *we obtain*

$$\mathbb{E}[\psi_{t+1} - \psi_t] \le -\mathbb{E}[\frac{\eta_x}{2}(\|\nabla\Phi(x_t)\|^2 - 3\epsilon^2)],$$

*and total oracle cost for finding stationary point is*

$$\#funtion = dT(pn + b) + dB = \mathcal{O}((d_1 + d_2)\sigma\epsilon^{-3}L_{\max}^4\kappa^8 + (d_1 + d_2)\sigma^2\epsilon^{-2}L_{\max}^2\kappa^4).$$

*Proof.* similar to finite sum case, the only difference with finite sum case is we have additional variance term

$$\Phi(x_{t+1}) \le \Phi(x_t) - \frac{\eta_x}{2}\|\nabla\Phi(x_t)\|^2 - \left(\frac{1}{2\eta_x} - \frac{L_\phi}{2}\right)\|x_{t+1} - x_t\|^2 + \frac{3\eta_x}{2}\|v_{x,t} - \hat{h}_{x,t}\|^2$$

$$+ \frac{3\eta_x}{2}((2L^2 + 4r_z^2\rho^2)\|y_t - y^*\|^2 + 4L^2\|z_t - z_t^*\|^2 + 2\delta^2 + 4r_z^2\rho\delta),$$

define Lyapunov function

$$\Psi_t := \Phi(x_t) + \frac{3\eta_x}{2p}\|v_{x,t} - \hat{h}_{x,t}\|^2 + \frac{72(L^2 + 2r_z^2\rho^2)\eta_x}{\eta_y\mu}\left(\|y_t - y_t^*\|^2\right) + \frac{144\eta_x L^2\rho^2}{\eta_z\mu}\left(\|z_t - z_t^*\|^2\right),$$

by the definition of $\Psi_t$ and lemmas E.6, E.12, E.18, E.20, we have

$$\mathbb{E}[\Psi_{t+1} - \Psi_t]$$

$$\le \mathbb{E}[-\frac{\eta_x}{2}\|\nabla\Phi(x_t)\|^2 - \left(\frac{1}{2\eta_x} - \frac{L_\phi}{2}\right)\|x_{t+1} - x_t\|^2 + \frac{3\eta_x}{2}(2\delta^2 + 4r_z^2\rho\delta) + \frac{\eta_x}{B}(9(1 + r_z^2)\sigma^2 + 72r_z^2\rho\delta + 18\delta^2)]$$

$$+ \mathbb{E}[\frac{36L^2\rho^2\eta_x}{\eta_z\mu}\left(-(1 - L\eta_z)\|z_{t+1} - z_t\|^2 + \frac{4}{\eta_z\mu}l_{Z^*}^2\|c_{t+1} - c_t\|^2 + \frac{4\eta_z}{\mu}(\left\|v_{z,t} - \hat{h}_{z,t}\right\|^2 + \left\|h_{z,t} - \hat{h}_{z,t}\right\|^2)\right)]$$

$$+ \mathbb{E}[\frac{18(L^2 + 2r_z^2\rho^2)\eta_x}{\eta_y\mu}\left(-(1 - 2L\eta_y)\|y_{t+1} - y_t\|^2 + \frac{4}{\eta_y\mu}l_{Y^*}^2\|x_{t+1} - x_t\|^2 + \frac{4\eta_y}{\mu}(\left\|v_{y,t} - \hat{h}_{y,t}\right\|^2 + \left\|h_{y,t} - \hat{h}_{y,t}\right\|^2)\right)]$$

$$+ \mathbb{E}[\frac{3(1 - p)\eta_x}{bp}\left((8r_z^2 + 3L^2)\|c_{t+1} - c_t\|^2 + 8L^2\|z_{t+1} - z_t\|^2 + 8r_z^2\rho\delta + 6\delta^2\right)],$$

use lemma E.11 and E.4 to bound $\left\|h_{y,t} - \hat{h}_{y,t}\right\|^2, \left\|h_{z,t} - \hat{h}_{z,t}\right\|^2$, let $\frac{1-p}{bp} = 1$, and rearrange the terms, we have

$$\mathbb{E}[\Psi_{t+1} - \Psi_t]$$

$$\le \mathbb{E}[-\frac{\eta_x}{2}\|\nabla\Phi(x_t)\|^2 - \left(\frac{1}{2\eta_x} - \frac{L_\phi}{2}\right)\|x_{t+1} - x_t\|^2 + \frac{9\eta_x(1 + r_z^2)\sigma^2}{B}]$$

$$+ \mathbb{E}[\eta_x\left(\left(21 + \frac{72(L^2 + 2r_z^2\rho^2)}{\eta_y\mu} + \frac{288L^2\rho^2}{\eta_z\mu} + \frac{18}{B}\right)\delta^2 + \left(60 + \frac{864L^2\rho^2}{\eta_z\mu} + \frac{72}{B}\right)r_z^2\rho\delta\right)]$$

$$+ \mathbb{E}[\frac{36L^2\rho^2\eta_x}{\eta_z\mu}\left(-(1 - L\eta_z)\|z_{t+1} - z_t\|^2 + \frac{4}{\eta_z\mu}l_{Z^*}^2\|c_{t+1} - c_t\|^2\right) + \frac{144L^2\rho^2\eta_x}{\mu^2}\left\|v_{z,t} - \hat{h}_{z,t}\right\|^2]$$

$$+ \mathbb{E}[\frac{18(L^2 + 2r_z^2\rho^2)\eta_x}{\eta_y\mu}\left(-(1 - 2L\eta_y)\|y_{t+1} - y_t\|^2 + \frac{4}{\eta_y\mu}l_{Y^*}^2\|x_{t+1} - x_t\|^2\right) + \frac{72(L^2 + 2r_z^2\rho^2)\eta_x}{\mu^2}\left\|v_{y,t} - \hat{h}_{y,t}\right\|^2]$$

$$+ \mathbb{E}[\frac{3(1 - p)\eta_x}{bp}\left((8r_z^2 + 3L^2)\|c_{t+1} - c_t\|^2 + 8L^2\|z_{t+1} - z_t\|^2\right)],$$

denote Lyapunov function $\psi_t = \Psi_t + \frac{72(L^2+2r_z^2\rho^2)\eta_x}{p\mu^2}\left\|v_{y,t}-\hat{h}_{y,t}\right\|^2 + \frac{144L^2\rho^2\eta_x}{p\mu^2}\left\|v_{z,t}-\hat{h}_{z,t}\right\|^2$,

and use lemma E.8 and E.14 to bound $\mathbb{E}[\left\|v_{y,t}-\hat{h}_{y,t}\right\|^2]$, $\mathbb{E}[\left\|v_{z,t}-\hat{h}_{z,t}\right\|^2]$, we have

$$\mathbb{E}[\psi_{t+1}-\psi_t]$$

$$\leq -\mathbb{E}[\frac{\eta_x}{2}\|\nabla\Phi(x_t)\|^2 - \left(\frac{1}{2\eta_x}-\frac{L_\phi}{2}\right)\|x_{t+1}-x_t\|^2 + \frac{\eta_x\sigma^2}{B}\left(9(1+r_z^2)+\frac{216(L^2+2r_z^2\rho^2)}{\mu^2}+\frac{864L^2\rho^2(1+r_z^2)}{\mu^2}\right)]$$

$$+\eta_x\left(21+\frac{72(L^2+2r_z^2\rho^2)}{\eta_y\mu}+\frac{288L^2\rho^2}{\eta_z\mu}+\frac{432(L^2+2r_z^2\rho^2)}{\mu^2}+\frac{864L^2\rho^2}{\mu^2}+\frac{216(L^2+2r_z^2\rho^2)}{\mu^2B}+\frac{1728L^2\rho^2}{\mu^2B}\right)\delta^2$$

$$+\eta_x\left(60+\frac{864L^2\rho^2}{\eta_z\mu}+\frac{2304L^2\rho^2}{\mu^2}+\frac{6912L^2\rho^2}{\mu^2B}\right)r_z^2\rho\delta$$

$$+\mathbb{E}[\frac{144L^2\rho^2\eta_x}{\eta_z\mu}\left(-(1-L\eta_z)\|z_{t+1}-z_t\|^2+\frac{4}{\eta_z\mu}l_{Z^*}^2\|c_{t+1}-c_t\|^2\right)]$$

$$+\mathbb{E}[\frac{72(L^2+2r_z^2\rho^2)\eta_x}{\eta_y\mu}\left(-(1-2L\eta_y)\|y_{t+1}-y_t\|^2+\frac{4}{\eta_y\mu}l_{Y^*}^2\|x_{t+1}-x_t\|^2\right)]$$

$$+\mathbb{E}[3\eta_x\left((8r_z^2+3L^2)\|c_{t+1}-c_t\|^2+8L^2\|z_{t+1}-z_t\|^2\right)]$$

$$+\mathbb{E}[\frac{72(L^2+2r_z^2\rho^2)\eta_x}{\mu^2}\left(3L^2\|c_{t+1}-c_t\|^2\right)+\frac{288L^2\rho^2\eta_x}{\mu^2}\left((8r_z^2+3L^2)\|c_{t+1}-c_t\|^2+8L^2\|z_{t+1}-z_t\|^2\right)],$$

rearrange terms we obtain:

$$\mathbb{E}[\psi_{t+1}-\psi_t]$$

$$\leq\mathbb{E}[-\frac{\eta_x}{2}\|\nabla\Phi(x_t)\|^2 - \left(\frac{1}{2\eta_x}-\frac{L_\phi}{2}\right)\|x_{t+1}-x_t\|^2 + \underbrace{\frac{\eta_x\sigma^2}{B}\left(9(1+r_z^2)+\frac{216(L^2+2r_z^2\rho^2)}{\mu^2}+\frac{864L^2\rho^2(1+r_z^2)}{\mu^2}\right)}_{C_B}]$$

$$+\eta_x\underbrace{\left(21+\frac{72(L^2+2r_z^2\rho^2)}{\eta_y\mu}+\frac{288L^2\rho^2}{\eta_z\mu}+\frac{432(L^2+2r_z^2\rho^2)}{\mu^2}+\frac{864L^2\rho^2}{\mu^2}+\frac{216(L^2+2r_z^2\rho^2)}{\mu^2B}+\frac{1728L^2\rho^2}{\mu^2B}\right)\delta^2}_{C_{\delta_1}}$$

$$+\eta_x\underbrace{\left(60+\frac{864L^2\rho^2}{\eta_z\mu}+\frac{2304L^2\rho^2}{\mu^2}+\frac{6912L^2\rho^2}{\mu^2B}\right)r_z^2\rho\delta}_{C_{\delta_2}}$$

$$-\mathbb{E}[\left(\frac{1}{2\eta_x}-\frac{L_\phi}{2}-\eta_x\left(\frac{288(L^2+2r_z^2\rho^2)l_{Y^*}^2}{\eta_y^2\mu^2}+\frac{576L^2\rho^2l_{Z^*}^2}{\eta_z^2\mu^2}\right)\right.$$

$$\left.-\eta_x\left(24r_z^2+9L^2+\frac{216(L^4+2L^2r_z^2\rho^2)}{\mu^2}+\frac{288L^2\rho^2(8r_z^2+3L^2)}{\mu^2}\right)\right)\|x_{t+1}-x_t\|^2]$$

$$-\mathbb{E}[\eta_x\left(\frac{144L^2\rho^2}{\eta_z\mu}-\frac{144L^3\rho^2}{\mu}-24L^2-\frac{2304L^4\rho^2}{\mu^2}\right)\|z_{t+1}-z_t\|^2]$$

$$-\mathbb{E}[\left(\frac{72(L^2+2r_z^2\rho^2)}{\eta_y\mu}-\frac{72(L^3+2Lr_z^2\rho^2)}{\mu}\right.$$

$$\left.-\eta_x\left(\frac{576L^2\rho^2}{\eta_z^2\mu^2}+(24r_z^2+9L^2)+\frac{216(L^4+2L^2r_z^2\rho^2)}{\mu^2}+\frac{288L^2\rho^2(8r_z^2+3L^2)}{\mu^2}\right)\right)\|y_{t+1}-y_t\|^2],$$

let's discuss the how to choose step size to let the coefficient of $\|x_{t+1}-x_t\|^2, \|y_{t+1}-y_t\|^2, \|z_{t+1}-z_t\|^2 \leq 0$, to simplify denote the coefficient of $\|x_{t+1}-x_t\|^2, \|y_{t+1}-y_t\|^2, \|z_{t+1}-z_t\|^2$ be $C_x, C_y, C_z$, we have

- $\|z_{t+1}-z_t\|^2$: let $\eta_z \leq \frac{144L^2\rho^2}{\mu(\frac{144L^3\rho^2}{\mu}+24L^2+\frac{2304L^4\rho^2}{\mu^2})} = \mathcal{O}(\frac{1}{L_{\max}\kappa})$, we obtain $C_z \leq 0$

- $\|y_{t+1} - y_t\|^2$: let $\eta_y \le \frac{1}{2\kappa}$, we obtain

$$C_y = -\mathcal{O}(\frac{L_{max}\kappa^3}{\eta_y} - \eta_x\mathcal{O}(L_{\max}^4\kappa^4))$$

we further let $\eta_y \le \frac{1}{\eta_x\mathcal{O}(L_{\max}^3\kappa)}$, thus we can obtain $C_y \le 0$.

- $\|x_{t+1} - x_t\|^2$: let $\eta_x = \frac{1}{2L_\phi}$, denote coefficient of $\|x_{t+1} - x_t\|^2$ as $C_x$, we have

$$C_x \le -\left(\frac{1}{4\eta_x} - \eta_x\mathcal{O}\left(\frac{\kappa^6}{\eta_y^2} + L^4\kappa^8\right)\right)$$

choose $\eta_x \le \min\{\frac{\eta_y}{L^{1.5}}, \frac{1}{L_{\max}^2\kappa^4}\}$, we have $C_x \le 0$, and we can easily validate that choosing $\eta_y = \frac{1}{2\kappa}, \eta_x = \mathcal{O}(\frac{1}{L^2\kappa^4})$ can simultaneously satisfy both $\eta_x \le \min\{\frac{\eta_y}{L_{\max}^{1.5}}, \frac{1}{L_{\max}^2\kappa^4}\}$ and $\eta_y \le \frac{1}{\eta_x\mathcal{O}(L_{\max}^3\kappa)}$.

since $C_x, C_y, C_z \le 0$, choose $\delta$ suffciently small,i.e, $\delta \le \frac{\epsilon^2}{\mathcal{O}(L_{\max}^5\kappa^4)}$ to let $C_{\delta_1} \le \frac{\eta_x\epsilon^2}{2}, C_{\delta_2} \le \frac{\eta_x\epsilon^2}{2}$, $B \ge \mathcal{O}(L_{\max}^2\kappa^4\epsilon^{-2}\sigma^2)$ to let $C_B \le \frac{\eta_x\epsilon^2}{2}$, we obtain:

$$\mathbb{E}[\psi_{t+1} - \psi_t] \le -\mathbb{E}[\frac{\eta_x}{2}(\|\nabla\Phi(x_t)\|^2 - 3\epsilon^2)],$$

thus $T = \frac{\epsilon^{-2}}{\eta_x} = \mathcal{O}(\epsilon^{-2}L_{\max}^2\kappa^4)$ and total oracle cost is

$$\#funtion = dT(pn + b) + dB = \mathcal{O}((d_1 + d_2)\sigma\epsilon^{-3}L_{\max}^4\kappa^8 + (d_1 + d_2)\sigma^2\epsilon^{-2}L_{\max}^2\kappa^4).$$

$\square$

