# OpenReview forum: "Provable Faster Zeroth-order Method for Bilevel Optimization with Optimal Dependency on Error and Dimension"
_ICLR.cc/2025/Conference — Submitted to ICLR 2025_

### Official Review · Reviewer_syFA · 2024-10-27

**Soundness:** 2
**Presentation:** 2
**Contribution:** 2
**Rating:** 6
**Confidence:** 5

**Summary:**

This paper considers bilevel optimization. In particular, the authors aim at solving the stochastic bilevel optimization via zeroth-order methods. Motivated by the fact that a recent work has high dimensional dependency in the complexity bound, the authors propose VRZSBO, and prove that it requires less oracle complexity than the existing work, in the context of zeroth-order bilevel optimization algorithms. Numerical simulations are provided to validate the performance of their algorithm under different settings and their better convergence as compared to existing algorithms.

**Strengths:**

1. The authors provide a better upper bound for the oracle complexity of zeroth-order bilevel optimization, without the need of additional assumptions.

2. The technical claims are in general sound.

**Weaknesses:**

1. Incomplete related work. Some important bilevel optimization algorithms are missing, even if they are not based on zeroth-order methods. It would be good if the authors could include a more comprehensive table containing existing algorithms with comparisons of assumptions, complexities, and types of oracles needed. See, for example, the related works in table 1 in [1].

2. Presentation.
    1. the authors claim one contribution is that they do not have additional assumptions like 3rd Lipschitz. However, this is not well reflected in Table 1.
    2. It seems that Section 1.1 can be moved to other sections for better presentation — for those who are familiar with bilevel optimization this section seems unnecessary and for those who are unfamiliar with bilevel optimization this section is full of technical details and hard to follow.
    3. It would be better if the authors could provide a well-organized Section 3 — the details of analysis described in 3.5 - 3.7 can be moved to the appendix to make the main context more readable. For example, proof sketch in section 3.6 is fairly standard and unnecessary to be included in the main body of the paper.

3. There are many important baselines missing in the experiments.

4. Overfull box in lines: 257 - 262, 409, 453, etc.

Reference

[1] BILEVEL OPTIMIZATION UNDER UNBOUNDED SMOOTHNESS: A NEW ALGORITHM AND CONVERGENCE ANALYSIS

**Questions:**

1. Can the authors explain the motivation of studying zeroth-order bilevel algorithms? The experiments can not reflect if VRZSBO can still be a better choice in large-scale settings.

2. Can the authors explain why (d_1 + d_2) dependency is optimal? For example, is there a reference for the lower bound of the dimension dependency.

3. The oracle cost mentioned in Remark 2 is in expectation (see (14) - (16)). However, the cost in (Aghasi and Ghadimi, 2024) is not in expectation, as the estimator therein does not require an additional sampling step (i.e., (14) - (16)). I am wondering if the authors could provide more context on this unfair comparison.

---

> ### Author Response · Authors · 2024-11-21
>
> We sincerely appreciate the reviewers' valuable suggestions regarding the writing of our paper, which have been very helpful in improving our work. Following your recommendations, we have revised the structure of the paper, included comparisons with more algorithms in the tables, and added additional experiment.
>
> **Q1: Can the authors explain the motivation of studying zeroth-order bilevel algorithms? The experiments can not reflect if VRZSBO can still be a better choice in large-scale settings**
>
> 1.Bilevel is a highly general framework that can model many real-world problems. Min-max problems and single-level optimization problems are both special cases of bilevel problems. Zeroth-order optimization has already been extensively studied in single-level and min-max problems, attracting significant interest. Therefore, studying zeroth-order algorithms under the bilevel framework to obtain more general conclusions compared to the aforementioned cases is highly appealing.
> 2.Moreover, reformulating some zeroth-order optimization tasks using the bilevel framework can potentially achieve better results. For example, as shown in [1] and [2], combining adversarial attacks (a common application of zeroth-order algorithms) with meta-learning (a common bilevel application) led to improved performance over existing methods.
> Given that [3] demonstrates the use of zeroth-order algorithms for fine-tuning large language models (LLMs), achieving up to 12× memory reduction and up to 2× GPU-hour reduction, and [4] highlights the close connection between the bilevel framework and LLM , zeroth-order bilevel algorithms hold significant potential for large-scale applications. Our VRZSBO algorithm, with its strong theoretical guarantees and low complexity, can serve as an important baseline for future research in zeroth-order bilevel optimization.
>
>
>
> **Q2  Can the authors explain why $(d _1 + d _2)$ dependency is optimal? For example, is there a reference for the lower bound of the dimension dependency.**
>
> Thank you for raising this question. We answer it in the General Response .
>
> **Q3: the cost in (Aghasi and Ghadimi, 2024) is not in expectation   while  our oracle cost  is in expectation**
>
> This problem is actually results in the difference between PAGE[5] and Spider[6].To achieve a single-loop algorithm, PAGE selects a large batch size with a probability of $p$, while SPIDER uses a large batch size once every $1/p$ iterations. This results in both methods having the same complexity, but PAGE's results are presented in an expected form. This reflects a trade-off between the single-loop structure and randomness. We do not consider this a significant issue, as in single-level problems, comparing the complexities of SPIDER and PAGE is not regarded as unfair.
>
> Reference：
>
> [1]Boosting Black-Box Adversarial Attacks with Meta Learning
>
> [2]Simulating Unknown Target Models for Query-Efficient Black-box Attacks
>
> [3]Fine-Tuning Language Models with Just Forward Passes
>
> [4]LLMS ARE HIGHLY-CONSTRAINED BIOPHYSICAL SE QUENCE OPTIMIZERS
>
> [5]PAGE: A Simple and Optimal Probabilistic Gradient Estimator for Nonconvex Optimization
>
> [6]SPIDER: Near-Optimal Non-Convex Optimization via Stochastic Path Integrated Differential Estimator

---

### Official Review · Reviewer_H91m · 2024-11-02

**Soundness:** 3
**Presentation:** 3
**Contribution:** 2
**Rating:** 6
**Confidence:** 3

**Summary:**

Using finite-difference approximations and the PAGE method for nonconvex optimization, this work introduces a novel zeroth-order algorithm, VRZSBO, for solving bilevel optimization problems with a strongly convex lower-level problem. The authors show that VRZSBO converges to an $\epsilon$-stationary point with a function query complexity of $\mathcal{O}((d_1 + d_2)\sqrt{n} \epsilon^{-2})$ in the finite-sum setting and $\mathcal{O}((d_1 + d_2) \epsilon^{-3})$ in the expectation setting, where $d_1$ and $d_2$ denote the dimensions of the upper- and lower-level variables, respectively. Experimental results on hyper-representation tasks using linear and two-layer network embedding models further validate the algorithm’s effectiveness.

**Strengths:**

$\textbf{S1:}$ The paper is easy to follow. Zeroth-order algorithms for bilevel optimization remain underexplored, even in the nonconvex-strongly-convex setting, making the introduction of the novel VRZSBO algorithm a valuable algorithmic contribution to this area.

$\textbf{S2:}$ The proposed VRZSBO algorithm achieves a significant  improvement over the best-known complexity of ZDSBA (Aghasi and Ghadimi, 2024), reducing the query complexity from $\mathcal{O}((d_1 + d_2)^4 \epsilon^{-3})$ to $\mathcal{O}((d_1 + d_2) \epsilon^{-3})$ in the expectation setting. Additionally, this work provides a complexity analysis for the finite-sum setting.

**Weaknesses:**

$\textbf{W1:}$ It is well-known that the Hessian-vector product can be implemented through finite-difference approximations of the gradient along a given direction. Given this, the novelty of the proposed results seems limited compared to existing methods.

Moreover, as noted in Chapter 9 of Numerical Optimization by Jorge Nocedal and Stephen J. Wright, finite-difference approaches are often impractical due to potential inaccuracies in function evaluations (e.g., rounding errors), which prevent the distance of finite differences from being arbitrarily small.

$\textbf{W2:}$ The experiments could be more comprehensive. Incorporating updated first-order algorithms, such as F2SA (Kwon et al., ICML 2023), would provide a more thorough comparison. Furthermore, including commonly used bilevel optimization tasks, such as hyperparameter selection and data hyper-cleaning, would further validate VRZSBO’s effectiveness, showcasing its reliability across various bilevel optimization problems.

I would be willing to increase the score based on the rebuttal.

**Questions:**

Apart from the questions raised in the Weaknesses section, some additional questions are as follows:

$\textbf{Q1:}$ Why does the proposed algorithm achieve an optimal result with respect to dependence on dimension? A bit more discussion is needed.

$\textbf{Q2:}$ In Assumption 2, why is there no stochastic assumption on $\nabla_{xy}^2 g$?

$\textbf{Q3:}$ Based on the proofs of Theorems 1 and 2, the iteration count $T$ in these theorems depends on $\psi_0$, which may in turn depend on $n$ (in the finite-sum case) and $\epsilon$ (in the expectation case). Therefore, to derive the complexity analysis results (i.e., the total oracle cost), does this require certain conditions on the initial points? Additionally, does it require an assumption of a lower bound for $\Phi(x)$? A bit more discussion on these points would be helpful.

$\textbf{Q4:}$ In Theorems 1 and 2, the zeroth-order tuning parameter $h$ does not appear—why is this the case? Is there a relationship between $v$ and $h$, potentially stemming from Corollary 1 and Lemma 3 through an intermediate parameter $\delta$? Additionally, in Figure 1, ablation studies examine the effects of different choices of $h$ and $v$ on the zeroth-order estimator. Were these experiments conducted with consideration of the relationship between $v$ and $h$?

$\textbf{Q5:}$ The theoretical results in this work suggest that smaller zeroth-order tuning parameters $v$ and $h$ yield better performance, yet the experimental results in Figure 1 do not support this conclusion. Could the authors explain this discrepancy?


$\textbf{Q6:}$ In the experiments, why was there no comparison with ZDSBA from Aghasi and Ghadimi (2024)?

$\textbf{Suggestions for improvement that did not affect the score:}$

In the convergence analysis, for example, in Theorem 1, it is stated that $\psi_{t+1}-\psi_{t} \leq -\frac{\eta_x}{2} ( | \nabla \Phi(x_t) |^2 + \epsilon^2)$. This implies that the Lyapunov function $\psi_t$ is decreasing. However, how is this possible when $\Phi(x)$ is nonconvex? It seems that the sign before $\epsilon^2$ may be incorrect.

In Theorem 1, why is there no $m$ in the finite-sum case? Is there an assumption that $m=\mathcal{O}(n)$?

Page 1, Line 045: In $g(x,y)$, $n$ should be $m$.

Page 2, Lines 102-103: Verify the form of $\bar{\nabla}\Phi(x)$.

Page 3, Line 145: Replace “$y(x)$, $y_\lambda(x)$, $\nabla \mathcal{L}\lambda(x)$” with “$y^*(x)$, $y\lambda^(x)$, $\nabla \mathcal{L}_\lambda^(x)$”.

Page 5: Line 228: The referenced equation (2) may not be appropriate.

Page 5: Lines 230-231: $R(x,y,z)$ does not include an inverse matrix operator.

Page 5, Lemma 1: It would be clearer to replace $y^$ with $y^(x_t)$ since $y^*$ is a function.

Page 5, Definition 1: In $\hat{\nabla}_x g(x,y)$, replace $d_2$ with $d_1$, and in $\hat{\nabla}_y f(x,y)$, replace $d_1$ with $d_2$.

Page 6, Lemma 3: Clarify the meaning of $r_z$ in the definition of $h$.

Page 7, Line 327: “Section 3.1” should be changed to “Sections 3.2–3.3”.

Page 7, Equations (14), (16): Keep the notations consistent with those in (10)–(11).

Page 7, Line 369: Should $-p$ replace $-\frac{1}{p}$ according to Lemma 9?

Page 7, Line 372:  Clarify what $\eta$ represents in the definition of $\Psi_t$.

Page 8, Lines 383-387: Should $-p$ replace $-\frac{1}{p}$ based on Lemmas 6–7?

Page 8, Line 398: Replace $\eta$ with $\eta_x$.

Page 8, Lemmas 4-5: Define $\ell_{Z^}$ and $\ell_{Y^}$.

Page 10, Figure 1: Correct typos, particularly in the caption.

---

> ### Author Response · Authors · 2024-11-21
> **Author Rebuttal  by Authors  (Part 1)**
>
> We sincerely thank you for taking the time to provide such detailed and thoughtful feedback. Your suggestions are helpful for improving our paper. We have supplemented the experiments and corrected the typos based on your comments in the revised version, and we will now address the another questions you raised.
>
> **W: finite-difference approaches are often impractical due to potential inaccuracies in function evaluations (e.g., rounding errors), which prevent the distance of finite differences from being arbitrarily small.**
>
> 1.To the best of our knowledge, achieving the best theoretical upper bound for a Hessian-free method appears challenging without employing finite differences. First-order methods based on penalty functions can at best achieve a complexity of $\mathcal{O}(\epsilon ^{-4})$ ([1], ICML 2024), whereas  finite-difference-based method achieves a significantly improved complexity of $\mathcal{O}(\epsilon ^{-3})$.
> 2.Although we did not employ this approach in our paper, we would like to introduce an intersting  method that can effectively address the issue of inaccuracies in function evaluations, provided that complex number computations are acceptable[1].
> Using a Taylor’s series expansion ( $e _{i}$ denote the $i$ th standard basis vector，v is a smoothing parameter, z is a vector):
>
> $$
> f(x+\mathrm{i} ve_{i})=f(x)+\mathrm{i} v \langle e_{i}, \nabla f(x)\rangle+\mathcal{O}(v ^2) \
> $$
> and
>
> $$
> \nabla f(x+\mathrm{i} v z)=\nabla f(x)+\mathrm{i}v \nabla ^2 f(x)z +\mathcal{O}(v ^2)
> $$
> we can estimate the gradient and hessian-vector product as :
> $\nabla f(x)=[Im(f(x+\mathrm{i} ve _{1})),Im(f(x+\mathrm{i} ve _{2})),\dots,Im(f(x+\mathrm{i} ve _{d}))] ^T$ and $\nabla f(x)z=Im(\nabla f(x+\mathrm{i} vz))$.
> This estimation method is not subject to rounding errors because it doesnt  involve subtraction,serves as a good complement to our approach, particularly when the estimation might be affected by rounding errors.
>
>  **Q: Why does the proposed algorithm achieve an optimal result with respect to dependence on dimension? A bit more discussion is needed.**
>
> Thank you for raising this question. We answer it in the General Response.
>
> **Q: In Assumption 2, why is there no stochastic assumption on $\nabla _{xy} ^2 g(x,y)$?**
>
> Sorry about this typo. We have corrected it.

---

> ### Author Response · Authors · 2024-11-21
> **Author Rebuttal  by Authors  (Part 2)**
>
> **Q: Based on the proofs of Theorems 1 and 2, the iteration count T in these theorems depends on $\psi _{0}$, which may in turn depend on n (in the finite-sum case) and ϵ (in the expectation case). Therefore, to derive the complexity analysis results (i.e., the total oracle cost), does this require certain conditions on the initial points? Additionally, does it require an assumption of a lower bound for Φ(x)? A bit more discussion on these points would be helpful.**
>
> Good question. recall that
>
> $$
>       \begin{align*}
>         \mathbb{E}[\psi _0]:&=\Phi(x _t)+\frac{18(L ^{2}+2 r _{z} ^{2}\rho ^{2})\eta _x}{\eta _y\mu}(|| y _{0}- y ^{\star} _{0}|| ^{2})+\frac{36\eta _x L ^{2} \rho ^{2}}{\eta _z\mu} (|| z _{0}- z ^{\star} _{0}|| ^{2})\\
>         &+\mathbb{E}[ \frac{3\eta _x}{2p}||v _{x, 0}-\hat{h} _{x, 0}|| ^{2}+ \frac{72(L ^{2}+2 r _{z} ^{2}\rho ^{2})\eta _x}{p\mu ^{2}}|| v _{y, 0} -\hat{h} _{y, 0} || ^{2}+ \frac{144 L ^{2} \rho ^{2}\eta _x}{p\mu ^{2}}|| v _{z, 0} -\hat{h} _{z, 0} || ^{2}],
> 	\end{align*}
> $$
> For the variance terms $\mathbb{E}[||v _{x, 0}-\hat{h} _{x, 0}|| ^{2}],\mathbb{E}[|| v _{y, 0} -\hat{h} _{y, 0} || ^{2}],\mathbb{E}[|| v _{z, 0} -\hat{h} _{z, 0} || ^{2}]$, by computeing
> $v _{z, 0}=\frac{1}{B}\sum _{i=1} ^{B} D\hat{H} _{v}(t+1;\xi _{i})-\hat{\nabla} _{y}f(x _{t+1}, y _{t+1};\zeta _{i})$, $v _{x,0}=\frac{1}{B}\sum _{i=1} ^{B}\hat{\nabla} _{x}f(x _{t+1}, y _{t+1};\zeta _{i})-D\hat{J} _{v}(t+1;\xi _{i})$, $v _{y,0}=\frac{1}{B}\sum _{i=1} ^{B}\hat{\nabla} _{y}g(x _{t+1}, y _{t+1};\xi _{i})$
> with batchsize $B=n$ (finite-sum case) and $\mathcal{O}(\epsilon ^{-2})$ (the expectation case) when $t=0$, the variance terms $\mathbb{E}[||v _{x, 0}-\hat{h} _{x, 0}|| ^{2}],\mathbb{E}[|| v _{y, 0} -\hat{h} _{y, 0} || ^{2}],\mathbb{E}[|| v _{z, 0} -\hat{h} _{z, 0} || ^{2}]$  will decrease to 0（finite-sum case）and $\mathcal{O}(1)$ （expectation case）, which makes $\mathbb{E}[\psi _0]$ independt of $n$ and $\epsilon$, we will add this initialization step to our algorithm.
> The remaining terms $\Phi(x _{0})$ , $|| y _{0}- y ^{\star}  _{0}|| ^{2}$  and  $|| z _{0}- z ^{\star} _{0}|| ^{2}$ , to the best of our knowledge, appear in almost all bilevel optimization literature. Since $y _{0} ^*$ and $z _{0} ^{\star}$ are solutions to strongly convex problems, we can run algorithm $\log(\epsilon)$ times to decrease $|| y _{0}- y ^{\star} _{0}|| ^{2}$ and $|| z _{0}- z ^{\star} _{0}|| ^{2}$ to nearly 0 .This trick is called  "warm start", which is  effective in certain experiments. And of course we require an assumption of a lower bound for $\Phi(x _{0})$, which is standard in bilevel optimization,  we will add this assumption    to improve clarity.
>
> **Q: In Theorems 1 and 2, the zeroth-order tuning parameter h does not appear—why is this the case? Is there a relationship between v and h, potentially stemming from Corollary 1 and Lemma 3 through an intermediate parameter δ? Additionally, in Figure 1, ablation studies examine the effects of different choices of h and v on the zeroth-order estimator. Were these experiments conducted with consideration of the relationship between v and h?**
>
> Your understanding is correct. In Theorems 1 and 2, $h$ does not explicitly appear because its effect is incorporated into $v$ through the intermediate parameter $\delta$, as shown in Corollary 1 and Lemma 3, we will include the value of $h$ in the statements of Theorems 1 and 2 to improve clarity.
> In the experiments, we fix v to the best-performing value, then test  h , and then swap by fixing h to the best-performing value and testing  v .
>
> **Q: The theoretical results in this work suggest that smaller zeroth-order tuning parameters v and h yield better performance, yet the experimental results in Figure 1 do not support this conclusion. Could the authors explain this discrepancy?**
>
> Good question. We would like to clarify that, theoretically, the value of  v indeed improves with smaller values. However, for h, there is an optimal value: it should not be too large or too small, as shown in the proof of Lemma D.6. In our experiments, the optimal value of h  corresponds to a middle-range value, which is consistent with our theoretical analysis. As for v , the experimental trend does not fully align with our theoretical expectations,  fortunately, we observed that the best performance in the experiments still corresponds to the smallest value of v. Based on our experience, selecting a sufficiently small v (e.g., 0.001 or 0.00001) remains effective.
>
> Reference：
>
> [1]On the Complexity of First-Order Methods in Stochastic Bilevel Optimization
>
> [2]Using Complex Variables to Estimate Derivatives of Real Functions

---

> > ### Comment · Reviewer_H91m · 2024-12-03
> > **Thanks for the response.**
> >
> > I thank the authors for their detailed responses and have adjusted my score accordingly.

---

> > > ### Author Response · Authors · 2024-12-03
> > >
> > > Thank you for reconsidering our work and for the updated score. We appreciate your valuable feedback and the time you invested in reviewing our paper.

---

### Official Review · Reviewer_L5fV · 2024-11-04

**Soundness:** 3
**Presentation:** 2
**Contribution:** 2
**Rating:** 3
**Confidence:** 3

**Summary:**

In this paper, the authors propose a zeroth-order method for solving stochastic bilevel optimization problems, where the lower-level problem is strongly convex and only function values are accessible for both levels. Their method can be regarded as a zeroth-order approximation of the SPABA method proposed by (Chu et al., 2024), where the gradients and Hessian-vector products are replaced by their zeroth-order estimators. Specifically, the authors employ a coordinate-wise zeroth-order estimator, approximating each coordinate of the gradient by finite differences. They analyze their proposed algorithm under both general stochastic and finite-sum settings, demonstrating that the function value query complexity scales linearly with the problem’s dimension and has optimal dependence on the target accuracy.

**Strengths:**

In the context of zeroth-order algorithms for bilevel optimization, the proposed algorithm improves dimensional dependence compared to prior work and introduces a simpler single-loop update format.

**Weaknesses:**

- My major concern is the use of the coordinate-wise zeroth-order estimator in Definition 1. As noted by the authors, this estimator requires $\mathcal{O}(d_1+d_2)$ function value queries in total, which may be as computationally expensive as obtaining the exact gradient by backpropagation. While it could be argued that only function values are accessible in certain applications, this does not hold for most common bilevel optimization applicationss, including the hyper-representation problem used in the experiments.
- In addition, the novelty of the paper appears somewhat limited. If my understanding is correct, the algorithmic framework closely follows the SPABA method by (Chu et al., 2024) such as the use of the PAGE variance reduction technique, and the main modidfication is to substitute the exact gradients and Hessian-vector products by their zeroth-order estimators. However, since the coordinate-wise zeroth-order estimator is employed, the gradient estimator error can theoretically be made arbitrarily small by selecting a sufficiently small $v$ (see Lemma 2). Indeed, the authors choose $v = \mathcal{O}(\epsilon^2)$ in Theorems 1 and 2, which means that the gradient error is also on the order of $\mathcal{O}(\epsilon^2)$. Consequently, the convergence analysis may not significantly differ from that of the original SPABA method.

----
Tianshu Chu, Dachuan Xu, Wei Yao, and Jin Zhang. SPABA: A single-loop and probabilistic stochastic bilevel algorithm achieving optimal sample complexity. ICML 2024

**Questions:**

- The author claims that the dependence on the dimension is optimal. While this appears reasonable, a rigorous lower bound is necessary to support this claim.
- The experiments should include the SPABA method as a baseline, as it is the most relevant algorithm for comparison with the proposed method. Additionally, comparisons across different algorithms should be based on runtime, given the potentially significant differences in per-iteration costs.

---

> ### Author Response · Authors · 2024-11-21
> **Author Rebuttal  by Authors  (Part 1)**
>
> We appreciate the reviewer’s thoughtful comments and feedback. In response to the concerns raised, we address the following points:
>
> **W: Concerning the use of the coordinate-wise zeroth-order estimator in Definition 1.**
>
> We understand the reviewer’s concern regarding the practical applicability of the coordinate-wise zeroth-order estimator. In response to this, we offer the following points:
>
> 1.We chose the coordinate-wise zeroth-order estimator in order to achieve stronger theoretical results.As discussed in our response to Reviewer Nm7C, the use of the coordinate-wise zeroth-order estimator enables us to achieve a complexity of $\mathcal{O}(d _{1}+d _{2})$. This result serves as an important baseline, as it matches the complexity of the most well-known zeroth-order min-max methods, which are often used in practice.
>
> 2. Early Stage of Zeroth-Order Bilevel Optimization Research
> We remind the reviewers that the first fully zeroth-order bilevel optimization algorithm was only recently proposed, so research in this area is still in its early stages. Given this, the improvement in our work from $(d _{1}+d _{2}) ^4$ to $d _{1}+d _{2}$ is a significant advancement.
> Our theoretical analysis provides valuable insights that future researchers can use to design methods tailored to specific problems. For example, a hypergradient estimator requiring only $\mathcal{O}(1)$ function evaluations or potentially even achieving faster speeds than first-order methods in practical tasks.
>
> 3. Moreover, reformulating some zeroth-order optimization tasks using the bilevel framework can potentially achieve better results. For example, as shown in [10] and [11], combining adversarial attacks (a common application of zeroth-order algorithms) with meta-learning (a common bilevel application) led to improved performance over existing methods.

---

> > ### Author Response · Authors · 2024-11-21
> > **Author Rebuttal  by Authors  (Part 2)**
> >
> > **W: Novelty of the paper**
> >
> > To address your concerns about novelty of the paper, we clarify them and improve discussion of challenges as follows：
> >
> > **1**. Analysis  of zeroth-order estimator
> > We would like to emphasize that, although in our analysis choosing a sufficiently small $v$ can indeed make the gradient error very small, the same cannot be done for the parameter $h$ used in Hessian-vector estimation. This is because, as shown in the proof of Lemma 2, there exists an optimal value for $h$: it should neither be too large nor too small. Our experiments on the optimal choice of $h$ also support this point. We believe that our results can effectively help avoid performance degradation caused by excessively small $h$, which might otherwise occur if one blindly minimizes $h$ without considering its optimal value.
> >
> > **2**.Even beyond the zeroth-order estimators, our approach to convergence analysis differs significantly from that of the SPABA method. Both methods are based on the quadratic auxiliary function framework proposed in [3], which typically divides the convergence analysis into three key components: (1) Descent Analysis of the Inner Problem, (2) Variance Reduction Analysis, and (3) Convergence Analysis of $\Phi(x)$. **Our analysis framework is novel and non-trivial** .Below, we compare our approach to SPABA in each of these aspects.
> >
> > (1) Descent Analysis of the Inner Problem
> > We now compar the difference between their descent in inner variable z (Lemma F.3) and ours (Lemma D.6).Although our analysis claims to leverage the strong convexity property, our proofs actually rely only on the Quadratic Growth (QG) condition for the objective function of $z$. In the unconstrained case, this condition is equivalent to the Polyak-Łojasiewicz (PL) condition [6]. In contrast, the proofs in SPABA require a property we refer to as co-coercivity, which applies specifically to strongly convex problems. Unfortunately, functions satisfying only the PL condition do not  exhibit the co-coercivity property. This distinction makes our analysis more general and potentially of broader interest, especially given the recent exploration of nonconvex-PL (NC-PL) bilevel problems [7],[8],[9].

---

> > > ### Author Response · Authors · 2024-11-21
> > > **Author Rebuttal  by Authors  (Part 3)**
> > >
> > > （2）Variance Reduction Analysis
> > > We now compar the difference between their variance descent in inner variable z (Lemma F.4) :
> > >
> > > $$
> > > \begin{align}
> > > &\mathbb{E}\left[\left||v _{k+1} ^z-D _z\left(x _{k+1}, y _{k+1}, z _{k+1}\right)\right|| ^2\right]  \\
> > > &\leq  \left(1-p+\frac{4(1-p)}{b}\left(L _2 ^g\right) ^2 \gamma _k ^2\right) \mathbb{E}\left[\left||v _k ^z-D _z\left(x _k, y _k, z _k\right)\right|| ^2\right] \\
> > > & +\left(2\left(L ^f\right) ^2+4 R ^2\left(L _2 ^g\right) ^2\right) \frac{(1-p)}{b} \alpha _k ^2 \mathbb{E}\left[\left||v _k ^x\right|| ^2\right] \\
> > > & +\left(2\left(L ^f\right) ^2+4 R ^2\left(L _2 ^g\right) ^2\right) \frac{(1-p)}{b} \beta _k ^2 \mathbb{E}\left[\left||v _k ^y-D _y\left(x _k, y _k, z _k\right)\right|| ^2\right] \\
> > > & +\left(2\left(L ^f\right) ^2+4 R ^2\left(L _2 ^g\right) ^2\right) \frac{(1-p)}{b} \beta _k ^2\left(L _1 ^g\right) ^2 \mathbb{E}\left[\left||y _k-y ^*\left(x _k\right)\right|| ^2\right] \\
> > > & +\frac{4(1-p)}{b}\left(L _2 ^g\right) ^2 \gamma _k ^2 L _z ^2 \mathbb{E}\left[\left||z _k-z ^*\left(x _k\right)\right|| ^2\right] \\
> > > & +\frac{4(1-p)}{b}\left(L _2 ^g\right) ^2 \gamma _k ^2 L _z ^2 \mathbb{E}\left[\left||y _k-y ^*\left(x _k\right)\right|| ^2\right]
> > > \end{align}
> > > $$
> > >
> > > while ours Lemma D.7 are:
> > > $$
> > >     \begin{align*}
> > >       &\mathbb{E} \left|| v _{z, t+1} -\hat{h} _{z, t+1} \right|| ^{2}\\
> > >       &\le (1-p)\mathbb{E}[\left||v _{z, t}- \hat{h} _{z, t}\right|| ^{2}]\\
> > >       &+\frac{2(1-p)}{b}((8 r _{z} ^{2}+3 L ^{2})\left|| c _{t+1}-c _{t} \right|| ^{2}\\
> > >       &+8L ^{2}\left|| z _{t+1}-z _{t} \right|| ^{2}+16 r _{z} ^{2} \rho \delta+6\delta ^{2}).
> > >   \end{align*}
> > > $$
> > > From a formal perspective, it is evident that our approach to error estimation differs significantly from SPABA, as the error terms we choose to analyze and utilize are substantially distinct.
> > >
> > > (3) Convergence Analysis of $\Phi(x)$.
> > > Although the potential function we use is similar in form to that of SPABA, our choice of coefficients is superior. For example, on page 40 of their paper, to ensure $\mathbb{E}\left[\left||z _k - z ^*\left(x _k\right)\right|| ^2\right]$ has a negative coefficient, the step size must satisfy $\alpha _k \leq \frac{\mu L _1 ^g \gamma _k}{12\left(\mu + L _1 ^g\right)\left(L _1 ^g\right) ^2}\left( i.e.\frac{\alpha _k}{\gamma _k}\leq  \frac{\mu  }{12\left(\mu + L _1 ^g\right)L _1 ^g}\right)$. On page 41, to ensure $\mathbb{E}\left[\left||v _k ^z - D _z\left(x _k, y _k, z _k\right)\right|| ^2\right]$ has a negative coefficient,the step size must satisfy $\gamma _k \leq \frac{\mu L _1 ^g}{12\left(\mu + L _1 ^g\right)} \alpha _k\left( i.e. \frac{\alpha _k}{\gamma _k}\geq \frac{ 12\left(\mu + L _1 ^g\right)}{\mu L _1 ^g} \right)$,a step size satisfying both conditions may not exist when $\mu ^2\leq 144(\mu+L _{1} ^g) ^2$, which is  highly likely to occur,however,in our potential function, the use of different weighting ratios can address this issue.
> > > Moreover, based on the reasons mentioned above, there is a significant difference between our approach and other works with near-optimal complexity, such as [4] and [5]. Therefore, we believe that our theoretical analysis is not merely a simple imitation of any single paper.
> > >
> > >
> > >
> > >
> > > **Q: The dependency on the dimension.**
> > >
> > > Thank you for raising this question. We answer it in the General Response.
> > >
> > > **Q: The experiments should include the SPABA method as a baseline, as it is the most relevant algorithm for comparison with the proposed method. Additionally, comparisons across different algorithms should be based on runtime, given the potentially significant differences in per-iteration costs.**
> > > We have added a new set of experiments as per your request, including SPABA as a baseline for comparison and reporting the runtime statistics.
> > >
> > > Reference：
> > >
> > > [1]Zeroth-Order Alternating Gradient Descent Ascent Algorithms for a Class of Nonconvex-Nonconcave Minimax Problems
> > >
> > > [2]Fully Zeroth-Order Bilevel Programming via Gaussian Smoothing
> > >
> > > [3]A framework for bilevel optimization that enables stochastic and global variance reduction algorithms
> > >
> > > [4]Achieving O(ϵ−1.5) Complexity in Hessian/Jacobian-free Stochastic Bilevel Optimization
> > >
> > > [5]A Lower Bound and a Near-Optimal Algorithm for Bilevel Empirical Risk Minimization
> > >
> > > [6]Linear Convergence of Gradient and Proximal-Gradient Methods Under the Polyak-Łojasiewicz Condition
> > >
> > > [7]Optimal Hessian/Jacobian-Free Nonconvex-PL Bilevel Optimization
> > >
> > > [8]On Penalty Methods for Nonconvex Bilevel Optimization and First-Order Stochastic Approximation
> > >
> > > [9]On Finding Small Hyper-Gradients in Bilevel Optimization: Hardness Results and Improved Analysis
> > >
> > > [10]Boosting Black-Box Adversarial Attacks with Meta Learning
> > >
> > > [11]Simulating Unknown Target Models for Query-Efficient Black-box Attacks

---

> ### Comment · Reviewer_L5fV · 2024-11-25
>
> I appreciate the authors' detailed response. However, I still have reservations regarding the use of the coordinate-wise zeroth-order estimator. This coordinate-wise finite-difference approach makes it not much different from a standard gradient oracle, in terms of both the computational cost and convergence analysis. While I understand that the authors aim to establish a baseline for zeroth-order methods in bilevel optimization, it is well-known that gradients can be well approximated using $O(d)$ function value queries, and thus a first-order method can be simulated using a zeroth-order oracle. In some sense, the contribution of this paper appears to be formalizing this observation for bilevel optimization problems.
>
> In addition, while there are some technical differences in the convergence analysis between this paper and SPABA, ultimately they appear to achieve similar convergence rates. It remains unclear whether any of these differences are essential for the algorithm to work properly with a coordinate-wise zeroth-order estimator.
>
> As a side note, I noticed that the two plots in Figure 3 appear to be identical. Intuitively, one would expect a larger dimension to result in a longer runtime. Could the authors double-check if this is correct?

---

> > ### Author Response · Authors · 2024-11-26
> >
> > Thank you for your thoughtful feedback. We appreciate the opportunity to address your concerns.
> >
> >  **Regarding the coordinate-wise zeroth-order estimator**
> >
> >  We reaffirm that our use of the coordinate-wise zeroth-order estimator aims to establish better theoretical convergence results. While we acknowledge that the coordinate-wise  approach is well-known, we argue that our specific application of this estimator to approximate the hypergradient in bilevel optimization introduces unique insights. For instance, as demonstrated in Lemma D.6 of our paper, the optimal choice of the parameter $h$ (step size for finite differences) is well-known: it cannot be too large or too small to ensure the correctness of the hypergradient estimation. This delicate balance is critical in the bilevel setting and is not a well-known result, adding a layer of novelty to our analysis. Additionally, our framework showcases how zeroth-order estimators can effectively handle the nested structure of bilevel problems without access to explicit gradients.
> >
> >
> >  **On convergence analysis and comparisons with SPABA**
> >
> > Regarding the technical differences in the convergence analysis between our paper and SPABA, we would like to clarify that, as discussed in our general response, the dependence of the convergence rates on $\epsilon$ in our work is already very close to optimal. Therefore, it is not surprising that our dependence on $\epsilon$ aligns with SPABA’s. However, the differences in the convergence analysis between our work and SPABA are indeed meaningful and carry practical implications, not only for zeroth-order methods but also for the analysis of first-order methods, as we have outlined in our previous responses.
> >
> > **Regarding the identical plots in Figure 3**
> >
> > We sincerely apologize for not updating the figure titles. The two plots in Figure 3 show the test loss and train loss, respectively. The experimental setup did not involve any change in the dimensionality of the problem. We will update the figure titles in the revised version.
> >
> >  We kindly ask you to reconsider the score in light of the clarifications and improvements made in the revised version. We are confident that our work provides meaningful contributions to the field of bilevel optimization, particularly in the context of zeroth-order methods.

---

### Official Review · Reviewer_Nm7C · 2024-11-04

**Soundness:** 3
**Presentation:** 2
**Contribution:** 2
**Rating:** 5
**Confidence:** 4

**Summary:**

This paper explores zeroth-order algorithms for addressing stochastic bilevel optimization problems. Expanding on earlier work that introduced a zeroth-order bilevel method using Gaussian smoothing, the authors enhance the dimensional dependency in sample complexity from $(d_1 + d_2)^4$ to $(d_1 + d_2)$. They also utilize variance reduction techniques to optimize the epsilon dependency in sample complexity from $\epsilon^{-4}$ to $\epsilon^{-3}$, achieving optimal bounds. The new algorithm undergoes theoretical analysis, demonstrating convergence to a stationary point with optimal complexity in both expectation and finite sum setting. Additionally, experiments are conducted to validate the algorithm's effectiveness.

**Strengths:**

The paper presents several key strengths that enhance its contribution to the field of stochastic bilevel optimization. Notably, it significantly improves the dimensional dependency in sample complexity from $(d_1 + d_2)^4$ to $(d_1 + d_2)$ and employs variance reduction techniques to optimize epsilon dependency, achieving optimal bounds. The thorough literature review effectively situates the work within existing research, underscoring its relevance. Additionally, the clear and structured presentation of the methodology supports reproducibility, while the empirical validation through experiments demonstrates the algorithm's practical effectiveness.

**Weaknesses:**

Lack of Novelty: The contributions of this paper build upon established concepts, integrating ideas from quadratic auxiliary functions, PAGE-type variance reduction, and zeroth-order gradient estimation. While these improvements are valuable, they may not be unexpected, in my humble opinion. For instance, ZDSBA built on previous work as a double-looped algorithm utilizing Hessian inverse approximation, so it is understandable that the incorporation of auxiliary quadratic functions would enhance dimension dependency. Additionally, the introduction of various variance reduction techniques is likely to improve epsilon dependency. Although I recognize the meaningful nature of these contributions, I feel that the paper does not fully meet the acceptance criteria for ICLR. Therefore, I will reserve my opinion on acceptance until after the rebuttal stage.

**Questions:**

Gaussian Smoothing vs. Coordinate-wise Smoothing: Based on my previous experiences, I have not observed a significant difference in the theoretical bounds between these two smoothing techniques. Consequently, I believe the improvements presented in the paper may not be attributed to coordinate-wise smoothing. The paper does not address the rationale behind choosing coordinate-wise smoothing over Gaussian smoothing. Could the authors please clarify the reasoning for this choice?

---

> ### Author Response · Authors · 2024-11-21
>
> Thank you for taking the time to review our paper. We appreciate the opportunity to clarify our contributions and provide a detailed comparison between our method and ZDSBA.
>
> **Q: Gaussian Smoothing vs. Coordinate-wise Smoothing**
>
> To simplify notation, let $d = d _1 + d _2$. The improved efficiency of our method with respect to dimensionality $d$ stems from two key factors: **(1)** Our single-loop structure eliminates the inner loop in [1], saving $\mathcal{O}(d)$ complexity from solving inner problems. **(2)** We use a finite-difference approximation for Hessian-vector or Jacobian-vector products, requiring only two gradient evaluations, with $\mathcal{O}(d)$ oracle cost. This is significantly more efficient than the Gaussian smoothing method in [1, Proposition 2.5], where approximating the Hessian-vector product incurs a variance of $\mathcal{O}(d ^3)$, leading to $\mathcal{O}(d ^3)$ complexity for hypergradient estimation and $\mathcal{O}(d ^3)$ outer loop iterations. Our method reduces this cost by $\mathcal{O}(d ^2)$. Combining the two above, our complexity is $d ^3$ faster than that of [1].
> 1. **Gradient Estimation:** Coordinate-wise smoothing requires $\mathcal{O}(d)$ oracle calls for accurate gradient estimation, while Gaussian smoothing uses $\mathcal{O}(1)$ calls but introduces a variance of $\mathcal{O}(d || \nabla f(x)  || ^2)$, ultimately requiring $\mathcal{O}(d)$ oracle calls to cancel this variance. Hence, the overall complexity for both remains $\mathcal{O}(d)$ times that of standard first-order methods.
> 2. **Hessian-Vector/Jacobian-Vector Estimation:** Finite-difference approximation via coordinate-wise smoothing requires $\mathcal{O}(d)$ oracle calls for accurate estimation, while [1, Proposition 2.5] requires $\mathcal{O}(1)$ calls but introduces a variance of $\mathcal{O}(d ^3)$ due to term $(d+4)(d+2)\left||\nabla _{x x} ^2 q\right|| _F ^2=\mathcal{O}(d ^3\left||\nabla _{x x} ^2 q\right|| _2 ^2)$, leading to $\mathcal{O}(d ^3)$ complexity in the outer loop. Our method reduces this by $\mathcal{O}(d ^2)$.
> **Conclusion:** Combining the $\mathcal{O}(d ^2)$ improvement in Hessian-vector/Jacobian-vector estimation and the $\mathcal{O}(d)$ speedup from the single-loop structure, our method achieves an overall $\mathcal{O}(d ^3)$ improvement compared to ZDSBA [1]. Additionally, our coordinate-wise results suggest the potential for Gaussian smoothing to achieve $\mathcal{O}(1)$ samples with $\mathcal{O}(d)$ variance, as both smoothing approaches perform similarly in gradient estimation, we have incorporated this comparison into the latest version of the paper.
>
> **W: Lack of Novelty**
> Thank you for your comment. We have already addressed a similar question raised by  Reviewer L5fV, and our response can be found in their section. We kindly refer you to that response for a detailed explanation, moreover, we would like to emphasize that our differences with SPABA also extend to comparisons with other methods that achieve near-optimal complexity [2],[3]. We believe that our answer should address your concern as well.
>
>
> Reference：
> [1]Fully Zeroth-Order Bilevel Programming via Gaussian Smoothing
>
> [2]Achieving O (ϵ−1.5) Complexity in Hessian/Jacobian-free Stochastic Bilevel Optimization
>
> [3]A Lower Bound and a Near-Optimal Algorithm for Bilevel Empirical Risk Minimization

---

### Author Response · Authors · 2024-11-21
**General Response**

We would like to thank all the reviewers for their thoughtful feedback and for raising important questions about our work. Several of these questions share common themes, and we have provided a general response below to address these shared concerns comprehensively.

**G: the optimality of $\epsilon$ and $d _{1},d _{2}$, and lowerbound.**
After we carefully check the lower bounds under our assumptions,we indeed find that our claim on "optimal" is not rigorous. We have decided to clarify the statements related to "optimal" to "best-known", and add a discussion on the dependencies on $\epsilon$ and $d$  in our revised version, based on the following facts:

1. Our dimension dependency $\mathcal{O}(d _{1} + d _{2})$ matches the best-known result in simpler  minmax problems [1, 2], and the $\epsilon$ dependency aligns with the best-known  complexity for first-order NC-SC bilevel problems [3, 4, 5], making our results "best-known."
2. Since [6] establishes a $\Omega(d)$ lower bound for single-level smooth convex problems, which are simpler than NC-SC bilevel problems (e.g., by setting $f(x, y) = f(x)$), a $\Omega(d _{1})$ dependency is inevitable.
3. Our assumptions align with those used to derive the lower bound in [4] for NC-SC  finite-sum bilevel problems, confirming that the $\mathcal{O}(\sqrt{n}\epsilon ^{-2})$ dependency is inevitable in the finite-sum case .
4. The $\mathcal{O}(\epsilon ^{-3})$ lower bound for single-level problems[8] in the expectation case is constructed under the mean-square-smooth assumption, which is slightly stronger than  the typical assumption of smoothness about smoothness of $f(x, y, \xi)$ for all $\xi$. Thus, strictly speaking, $O(\epsilon ^{-3})$ is not necessarily optimal in $\epsilon$ for the expectation case.

Based on existing lower bounds, the dependency on $d _1$ is inevitable. However, the optimality of the $d _2$ dependency remains unclear. It would be particularly interesting if the exponent of $d _2$ could be reduced to less than 1 for lower level strongly convex optimization, as this would imply a theoretical speedup of zeroth-order algorithms compared to first-order algorithms (assuming the gradient computation cost is $\mathcal{O}(d _1 + d _2)$).



[1]Zeroth-order algorithms for nonconvex–strongly-concave minimax problems with improved complexities

[2]Zeroth-Order Alternating Gradient Descent Ascent Algorithms for a Class of Nonconvex-Nonconcave Minimax Problems

[3]Achieving O (ϵ−1.5) Complexity in Hessian/Jacobian-free Stochastic Bilevel Optimization

[4]A Lower Bound and a Near-Optimal Algorithm for Bilevel Empirical Risk Minimization

[5]SPABA: A Single-Loop and Probabilistic Stochastic Bilevel Algorithm Achieving Optimal Sample Complexity

[6]Optimal Rates for Zero-Order Convex Optimization: The Power of Two Function Evaluations

[7]On the Complexity of First-Order Methods in Stochastic Bilevel Optimization

[8]Lower bounds for non-convex stochastic optimization

---

### Meta-Review · Area_Chair_YRe5 · 2024-12-14

**Metareview:**

This paper studies zeroth-order method for bilevel optimization. In particular, the authors proposed a single-loop accelerated zeroth-order algorithm for bilevel optimization. The reviewers believe that the proposed algorithm closely follows some existing works, and the novelty is limited. Moreover, it requires significant amount of work to improve the presentation and readability of the paper.

**Additional Comments On Reviewer Discussion:**

Further discussed the novelty and numerical experiments. The reviewers were not convinced.

---

### Decision · Program_Chairs · 2025-01-22

Reject